# Mamo decodes hierarchical temporal gradients into terminal neuronal fate

Ling-Yu Liu[1], Xi Long[1], Ching-Po Yang[1], Rosa L Miyares[1], Ken Sugino[1], Robert H Singer[1,2,3], Tzumin Lee[1]*

[1]Howard Hughes Medical Institute, Janelia Research Campus, Ashburn, United States; [2]Department of Anatomy and Structural Biology, Gruss Lipper Biophotonics Center, Albert Einstein College of Medicine, New York, United States; [3]Dominick P Purpura Department of Neuroscience, Gruss Lipper Biophotonics Center, Albert Einstein College of Medicine, New York, United States

**Abstract** Temporal patterning is a seminal method of expanding neuronal diversity. Here we unravel a mechanism decoding neural stem cell temporal gene expression and transforming it into discrete neuronal fates. This mechanism is characterized by hierarchical gene expression. First, *Drosophila* neuroblasts express opposing temporal gradients of RNA-binding proteins, Imp and Syp. These proteins promote or inhibit *chinmo* translation, yielding a descending neuronal gradient. Together, first and second-layer temporal factors define a temporal expression window of BTB-zinc finger nuclear protein, Mamo. The precise temporal induction of Mamo is achieved via both transcriptional and post-transcriptional regulation. Finally, Mamo is essential for the temporally defined, terminal identity of $\alpha'/\beta'$ mushroom body neurons and identity maintenance. We describe a straightforward paradigm of temporal fate specification where diverse neuronal fates are defined via integrating multiple layers of gene regulation. The neurodevelopmental roles of orthologous/related mammalian genes suggest a fundamental conservation of this mechanism in brain development.

DOI: https://doi.org/10.7554/eLife.48056.001

*For correspondence:
leet@janelia.hhmi.org

## Introduction

The brain is a complicated organ which not only requires specific connections between neurons to form circuits, but also many neuronal types with variations in morphology, neurotransmitters and receptors. While mechanisms controlling neuronal diversity have not been globally examined, studying neural stem cells in the mouse and fruit fly have given insight into key aspects of neuronal specification. For example, in the mouse neocortex, radial glial progenitors (RGP) are multipotent—they produce a variety of neuron types organized sequentially into six layers, and then produce glia (*Adnani et al., 2018*). In vivo lineage analysis demonstrated that after a stage of symmetric cell division, an individual neurogenic RGP produces an average of 8–9 progeny (range of 3–16) that can span all cortical layers (*Gao et al., 2014*). In *Drosophila*, clonal analysis has demonstrated a vast range of stem cell-specific lineage programs (*Ito et al., 2013*; *Yu et al., 2013*). On one extreme, lineage tracing of a single antennal lobe (AL) stem cell revealed a remarkable series of 40 morphologically distinct neuronal types generated sequentially (*Lin et al., 2012*; *Yu et al., 2010*). In light of these observations, a fundamental goal is to understand how distinct neuronal types correctly differentiate from a single progenitor. Despite a fundamental role for temporal patterning to create diverse neuronal lineages, our understanding of neuronal temporal patterning is still limited. While scientists have discovered key temporal factors expressed in neural progenitors, much less is understood about how these signals are interpreted, that is what factors lie downstream of the specification signals to determine distinct neuronal temporal fates.

Despite its relatively small brain, *Drosophila* is leading the charge on studies of neuronal temporal fate specification (*Courgeon and Desplan, 2019*; *Doe, 2017*; *Miyares and Lee, 2019*). Many temporal transcription factors originally discovered in the fly have since been confirmed to have conserved roles in mouse retinal and cortical development (*Holguera and Desplan, 2018*). Moreover, temporal expression of an RNA binding protein, IGF-II mRNA-binding protein (Imp), that guides temporal patterning in the postembryonic fly brain (*Liu et al., 2015*) is also implicated in mouse brain development (*Nishino et al., 2013*). *Drosophila* brain development is an excellent model for studying neurogenesis; the neural stem cells, called neuroblasts (NB), are fixed in number, their modes of division are well characterized, and each NB produces a distinctive series of neurons which change fate based on birth order (*Yu et al., 2013*). Finally, the fruit fly is a genetically tractable system making it ideal for studying gene networks involved in cell fate decisions.

In *Drosophila*, cycling NBs express age-dependent genes that provide the serially derived newborn neurons with different temporal factors. In the embryonic ventral nerve cord and the optic lobe, the NBs express a rapidly changing series of four to six temporal transcription factors (tTF), some of which are directly inherited by the daughter neurons (*Baumgardt et al., 2009*; *Isshiki et al., 2001*; *Kanai et al., 2005*; *Li et al., 2013*). Each tTF directly acts to specify a small number (two to four) of neuronal progeny. The neuronal progeny produced from one tTF window to the next can be quite different. The tTF series are intrinsically controlled, which ensures reliable production of all neuron types, but lacks the ability to adapt to complicated or changing conditions.

A separate mechanism is therefore required for adult brain development—both to produce very long series of related neuronal types and to coordinate with organism development. This can be accomplished utilizing protein gradients and hierarchical gene regulation, such as the mechanism used to pattern the fly's anterior/posterior (A/P) axis (*Rivera-Pomar et al., 1995*; *Rivera-Pomar and Jäckle, 1996*; *Struhl et al., 1989*; *Wang and Lehmann, 1991*). In *Drosophila* A/P patterning, the embryo is progressively partitioned into smaller and smaller domains through layered gene regulation. This is initiated by asymmetric localization maternal mRNAs, *bicoid* (anterior) and *nanos* (posterior). The resulting opposing proteins gradients then act on maternal mRNA translation, and in the case of Bicoid, zygotic transcription. The embryo then progresses through expression of maternal morphogen gradients, then zygotic expression of gap genes to determine broad embryo regions, followed by progressive segmentation by the pair-rule and segment polarity genes, and finally specification by the homeotic selector genes.

Notably, in postembryonic brain development, we have discovered two proteins in opposing temporal gradients expressed in NBs. These proteins are Imp and Syncrip (Syp) RNA-binding proteins. Imp and Syp control neuronal temporal fate specification as well as the timing of NB termination (decommissioning; *Liu et al., 2015*; *Ren et al., 2017*; *Syed et al., 2017*; *Yang et al., 2017*). Imp promotes and Syp inhibits translation of the BTB-zinc finger nuclear protein, *chinmo* (chronologically inappropriate morphogenesis), so that protein levels in newborn neurons descend over time (*Figure 1A*) (*Liu et al., 2015*). The level of Chinmo correlates with the specification of multiple neuronal temporal fates (*Zhu et al., 2006*). Discovering downstream layers in the Imp/Syp/Chinmo hierarchy is essential to fully comprehend the intricacies of temporal patterning in brain development.

Temporal regulation in the fly brain is easily studied in the relatively simple mushroom body (MB) neuronal lineages which are comprised of only three major cell types. These neuronal types are born in sequential order: beginning with γ neurons, followed by α′/β′ neurons, and finally α/β neurons (*Ito et al., 1997*; *Lee et al., 1999*). Imp and Syp are expressed in relatively shallow, opposing temporal gradients in the MB NBs. Modulation of Imp or Syp expression results in shifts in the neuronal temporal fate. Imp and Syp post-transcriptionally control Chinmo so that it is expressed in a gradient in the first two temporal windows (*Liu et al., 2015*). γ neurons are produced in a high Chinmo window, α′/β′ neurons are produced in a low Chinmo window, and α/β neurons are produced in a window absent of Chinmo expression (*Zhu et al., 2006*). Moreover, altering Chinmo levels can shift the temporal fate of MB neurons accordingly, strongly implicating dose-dependent actions, similar to that of a morphogen. Despite its importance in temporal patterning, the mechanisms underlying the dosage-dependent effects of Chinmo on neuronal temporal identity is unknown.

Here we describe Mamo (maternal gene required for meiosis, *Mukai et al., 2007*), a BTB zinc finger transcription factor critical for temporal specification of α′/β′ neurons. Mamo is expressed in a low Chinmo temporal window and Mamo expression can be inhibited both by high Chinmo levels and loss of *chinmo*. Additionally, Mamo is post-transcriptionally regulated by the Syp RNA binding

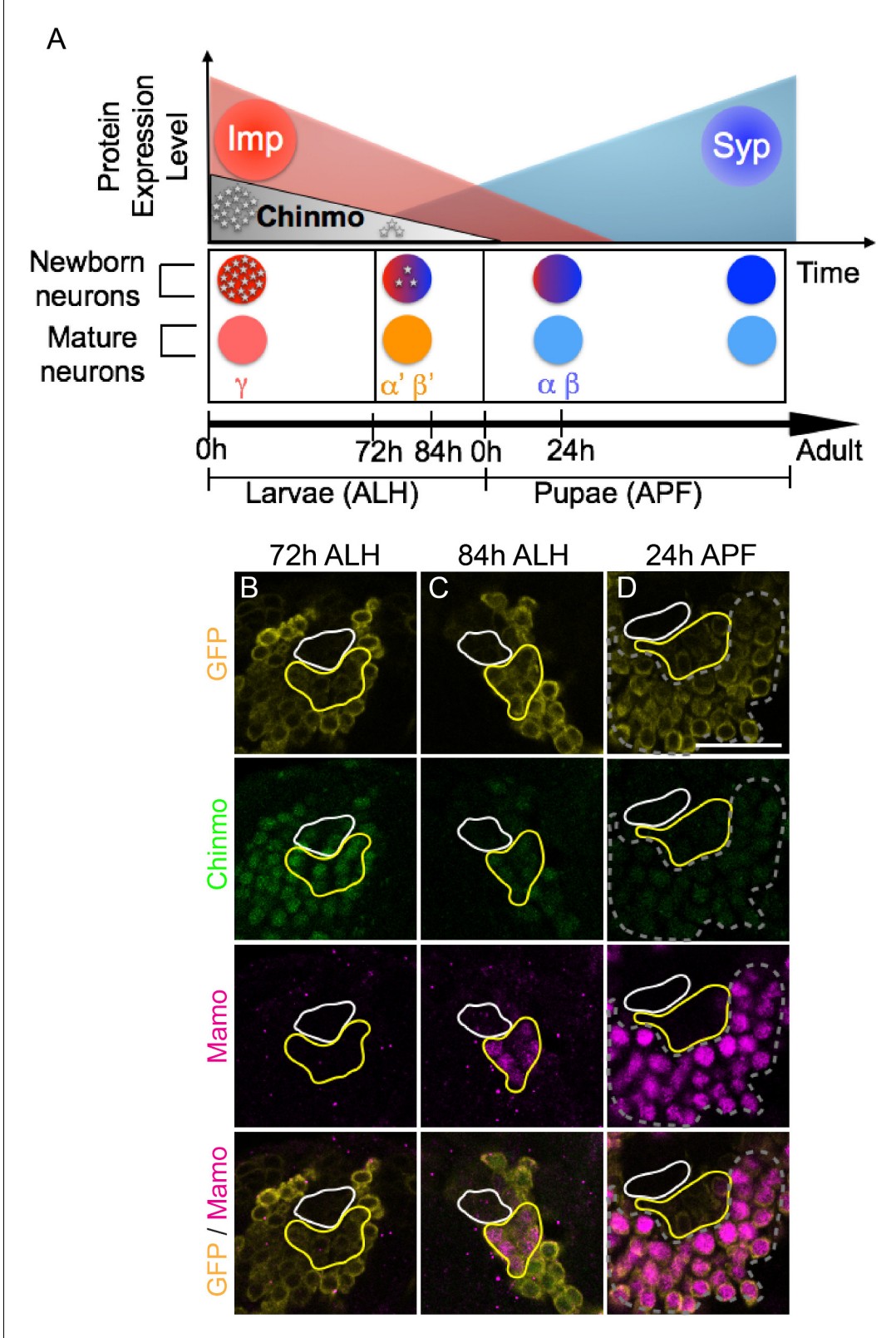

**Figure 1.** Mamo expression coincides with the generation of α′/β′ neurons in the Mushroom Body (MB) neuronal lineages. (**A**) Temporal gradients specify postembryonic neurons of the MB lineages into three sequential neuronal classes (*Lee et al., 1999*; *Liu et al., 2015*). Newborn neurons are colored to illustrate expression levels of Imp (red), Syp (blue), and Chinmo (gray stars). ALH = after larval hatching, APF = after pupal formation. (**B–D**) MB lineages (*OK107 > GFP*) immunostained for GFP, Chinmo (Rat-anti-Chinmo), and Mamo at different developmental times. A single focal plane near
*Figure 1 continued on next page*

*Figure 1 continued*

the MB NB is shown. Newborn neurons (NN) are identified by the very dim GFP expression near the NB as described by *Zhu et al. (2006)* and outlined in white. Young/maturing neurons are immediately adjacent to the NNs with a slightly higher GFP intensity and outlined in yellow. Chinmo levels in NNs decline over time. Mamo staining is visible at 84 hr ALH in young/maturing neurons (C). At 24 hr APF, Mamo expression is strong in older neurons (gray dashed outline), but absent from young/maturing neurons (D). Scale bar = 20 μm. Images are representative of n > 18. The quantification of Chinmo and Mamo staining is in *Figure 1—figure supplement 1*.

DOI: https://doi.org/10.7554/eLife.48056.002

The following source data and figure supplement are available for figure 1:

**Source data 1.** Intensity of Chinmo and Mamo staining at different developmental times.
DOI: https://doi.org/10.7554/eLife.48056.004
**Figure supplement 1.** Mamo expression coincides with the generation of α′/β′ neurons in the Mushroom Body (MB) neuronal lineages.
DOI: https://doi.org/10.7554/eLife.48056.003

protein. This layered regulation, which is utilized in both MB and AL lineages results in a discrete window of Mamo expression in young, post-mitotic neurons. In the MB lineages, this window corresponds to the middle window of neurogenesis and we establish that Mamo codes for middle temporal fate(s); α′/β′ neuronal characteristics are lost when Mamo levels are reduced and ectopic Mamo drives an increase in α′/β′ neuron production. The temporal fate determination paradigm we describe utilizes multiple levels of gene regulation. Temporal fate specification begins in the stem cell and proceeds in a hierarchical manner in successive stages where top and second-tier factors work together to specify neuronal temporal fate. Our data suggest that Mamo deciphers the upstream temporal specification code and acts as a terminal selector to determine neuronal fate.

## Results

### Mamo expression coincides with generation of α′/β′ neurons in the MB lineages

In order to understand how the descending Chinmo protein gradient could result in distinct temporal windows, we set out in search of potential Chinmo target genes. We identified Mamo as a candidate based on its expression pattern in the developing MB lineages. Mamo expression seems to trail weak Chinmo expression in both time and space (*Figure 1B–D*). α′/β′ neurons are specified in a temporal window when Chinmo levels in newborn neurons are weak, beginning around 72 hr after larval hatching (h ALH) (*Zhu et al., 2006*) (*Figure 1B*, *Figure 1—figure supplement 1A*). Mamo's expression is initiated a few hours afterwards (around 84 hr ALH) in a group of neurons that border the newborn neurons (*Figure 1C*, *Figure 1—figure supplement 1A*). This group of neurons is discernible by intermediate GFP levels driven by OK107-Gal4, and is hereafter referred to as young/maturing neurons. These data suggest that as the weak Chinmo expressing newborn neurons mature and move further from the NB, they begin to express Mamo. Conversely, Mamo is undetectable in young/maturing neurons that are destined to become γ (*Figure 1B*) or α/β neurons (*Figure 1D*). To validate that at 84 hr ALH, Mamo is in fact expressed in prospective α′/β′ rather than γ neurons, we used a γ neuron-specific driver and confirmed that there is no overlap with Mamo expression (*Figure 1—figure supplement 1B*). These results indicate a temporal induction of Mamo specifically in the prospective α′/β′ neurons, consistent with Mamo being a target of weak Chinmo within the young neuron stage of neuronal maturation. γ neurons, which express high Chinmo in early larval stages, begin to express Mamo during puparium formation (*Figure 1—figure supplement 1C*).

### Weak Chinmo initiates Mamo protein expression

To test if Mamo lies downstream of weak Chinmo, we examined the effect of altering Chinmo levels on Mamo expression. Consistent with our hypothesis, both overexpressing and eliminating Chinmo effectively abolished Mamo expression (*Figure 2B and C*, *Figure 2—figure supplement 1A*). Chinmo's effect on Mamo expression appears to be cell autonomous, as wild type neurons adjacent to *chinmo* null MARCM clones continue to express Mamo (*Figure 2C*, yellow arrow). Conversely, targeted *chinmo* RNAi prematurely reduced, rather than eliminating Chinmo (*Figure 3—figure supplement 1B&G*). This premature reduction in Chinmo initiated early Mamo expression (*Figure 2F*,

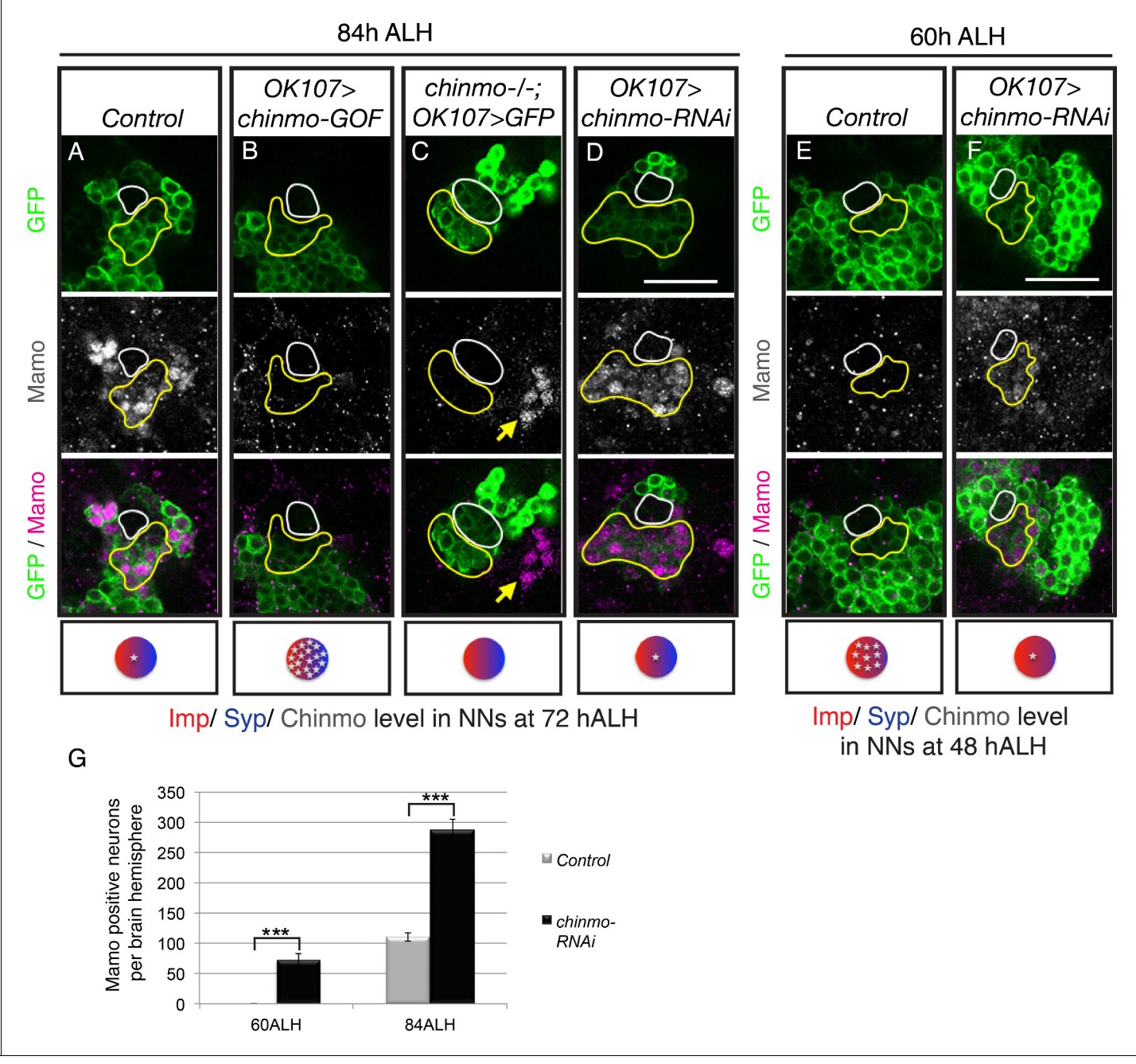

**Figure 2.** Weak Chinmo initiates Mamo protein expression. MB lineages (*OK107 > GFP*) with different genetic manipulations immunostained for GFP and Mamo. A single focal plane near the MB NB is shown. Newborn neurons (NN) are outlined in white. Young/maturing neurons are outlined in yellow. Images are representative of n > 18. Scale bar = 20 μm. The diagram below shows approximate levels of Imp (red), Syp (blue), and Chinmo (stars) expressed in the young/maturing neurons when they were NNs ~ 12 hr prior (as reported by *Liu et al., 2015*), or *Figure 3—figure supplement 1*). (**A-F**) At 84 hr ALH, Mamo staining is visible only in the young/maturing neurons in control brains (**A**) and brains expressing *chinmo-RNAi* (**D**). (**C**) *chinmo⁻/⁻; OK107 > GFP* is a *chinmo* null MARCM clone induced at newly hatched larvae (NHL). Note that *OK107* drives GFP only within the clone. MB neurons outside of the *chinmo⁻/⁻* clone (eyeless⁺, data not shown) express Mamo (yellow arrow). (**F**) At 60 hr ALH, Mamo staining is only visible after OK107 > *chinmo RNAi*. (**G**) Mean number (± SEM) of Mamo positive neurons per brain hemisphere in control (gray) and *chinmo-RNAi* (black) expressing MBs (***p<0.005, n = 4–5). The quantification of Mamo staining is in *Figure 2—figure supplement 1*.

DOI: https://doi.org/10.7554/eLife.48056.005

The following source data and figure supplement are available for figure 2:

**Source data 1.** Quantification of Mamo positive neurons.

DOI: https://doi.org/10.7554/eLife.48056.007

*Figure 2 continued on next page*

*Figure 2 continued*

**Source data 2.** Mamo staining intensity in young/maturing neurons with chinmo manipulations.
DOI: https://doi.org/10.7554/eLife.48056.008
**Figure supplement 1.** The intensity quantification of Mamo staining in different manipulations of MB neurons.
DOI: https://doi.org/10.7554/eLife.48056.006

*Figure 2—figure supplement 1B*) and thus greatly expanded (approximately 3-fold) the number of cells expressing Mamo at 84 hr ALH (*Figure 2D and G*). These results indicate that weak Chinmo expression is both necessary and sufficient to activate Mamo in young/maturing MB neurons.

## Mamo requires post-transcriptional regulation by Syp

*chinmo-RNAi* effectively reduced Chinmo levels early in development, so that immunostaining at 48 hr ALH reveals very low levels, both in newborn neurons and in older, early-born neurons (*Figure 3— figure supplement 1B and G*). Nevertheless, the initiation of Mamo expression is only shifted as early as 60 hr ALH (*Figure 2F*, *Figure 2—figure supplement 1B*, *Figure 3—figure supplement 2A and B*). This timing correlates with the onset of Syp expression (*Liu et al., 2015*), making us wonder whether Mamo could also be regulated by upstream temporal factor Syp. This would be analogous to how the pair-rule gene expression in a particular stripe is controlled by both maternal gradients and subsequent gap gene expression (*Small et al., 1991*). We therefore monitored Chinmo and Mamo expression following manipulations of Syp (*Figure 3* and *Figure 3—figure supplements 1 and 2C*). Repressing Syp increased Chinmo expression (*Figure 3—figure supplement 1E and G*) and thus abolished Mamo expression at 84 hr ALH (*Figure 3B*). Ectopic Syp marginally reduced Chinmo levels (*Figure 3—figure supplement 1F*) and there was a concomitant shift in Mamo expression—like *chinmo-RNAi,* there was an increased number of Mamo-positive cells at 84 hr ALH (*Figure 3C*). It was not clear whether this result was due solely to reduced Chinmo levels or a potential role for Syp in Mamo expression. Hence, we needed to create a scenario where Syp levels and Chinmo levels were uncoupled. We first used *Syp-RNAi* to remove Syp, which positively regulates Chinmo (*Figure 3—figure supplement 1E*) and then added *chinmo-RNAi* to lower the Chinmo levels (*Figure 3—figure supplement 1C and G*). Intriguingly in this scenario, Mamo was still absent (*Figure 3D*). Without Syp, weak Chinmo was no longer sufficient to promote Mamo expression.

Syp has been shown to regulate mRNA stability and translation (*McDermott et al., 2012*). It is therefore possible that *mamo* is transcriptionally controlled by weak Chinmo but post-transcriptionally regulated by the Syp RNA-binding protein. To differentiate transcription and post-transcriptional mRNA regulation, we turned to single molecule fluorescent in-situ hybridization (smFISH). We monitored the expression of both nascent and mature *mamo* transcripts with differentially labeled intron and exon probes (*Long et al., 2017*) (*Figure 4A* and *Figure 4—source data 2*). Bright nuclear foci of nascent transcripts indicate a site of active transcription (*Figure 4B*). We detected an onset of *mamo* transcription in the nuclei of newborn MB neurons starting at 72 hr ALH (*Figure 4D*). Mature *mamo* transcripts then gradually accumulated in the cytoplasm of young/maturing neurons (*Figure 4E*). Consistent with our previous results, knocking down Chinmo by targeted RNAi elicited a precocious activation of *mamo* transcription as early as 48 hr ALH (*Figure 4F*). When examining MBs lacking Syp that also had very weak Chinmo expression (*Syp-RNAi + chinmo-RNAi*, *Figure 3— figure supplement 1E and I*), we found sites of active *mamo* transcription at 48 hr ALH (*Figure 4I*). This clearly illustrates that repressing *Syp* did not delay the precocious induction of *mamo* transcription due to *chinmo-RNAi*. Instead, loss of Syp blocked the accumulation of mature *mamo* transcripts (*Figure 4I–K* and *Figure 4—figure supplement 1*). Meanwhile, without Syp the active sites of *mamo* transcription were short-lived, never surviving beyond the newborn neuron stage (*Figure 4I–K* and *Figure 4—figure supplement 1*). These observations indicate that Syp is required post-transcriptionally for *mamo* mRNA maturation and sustained *mamo* transcription. This transcriptional maintenance may be due to positive feedback by the Mamo protein itself. Consistent with this notion, *mamo-RNAi* did not inhibit *mamo* induction in newborn neurons (*Figure 4L* and *Figure 4—figure supplement 1*), but prevented *mamo* mRNA maturation and sustained *mamo* transcription (*Figure 4M* and *Figure 4—figure supplement 1*).

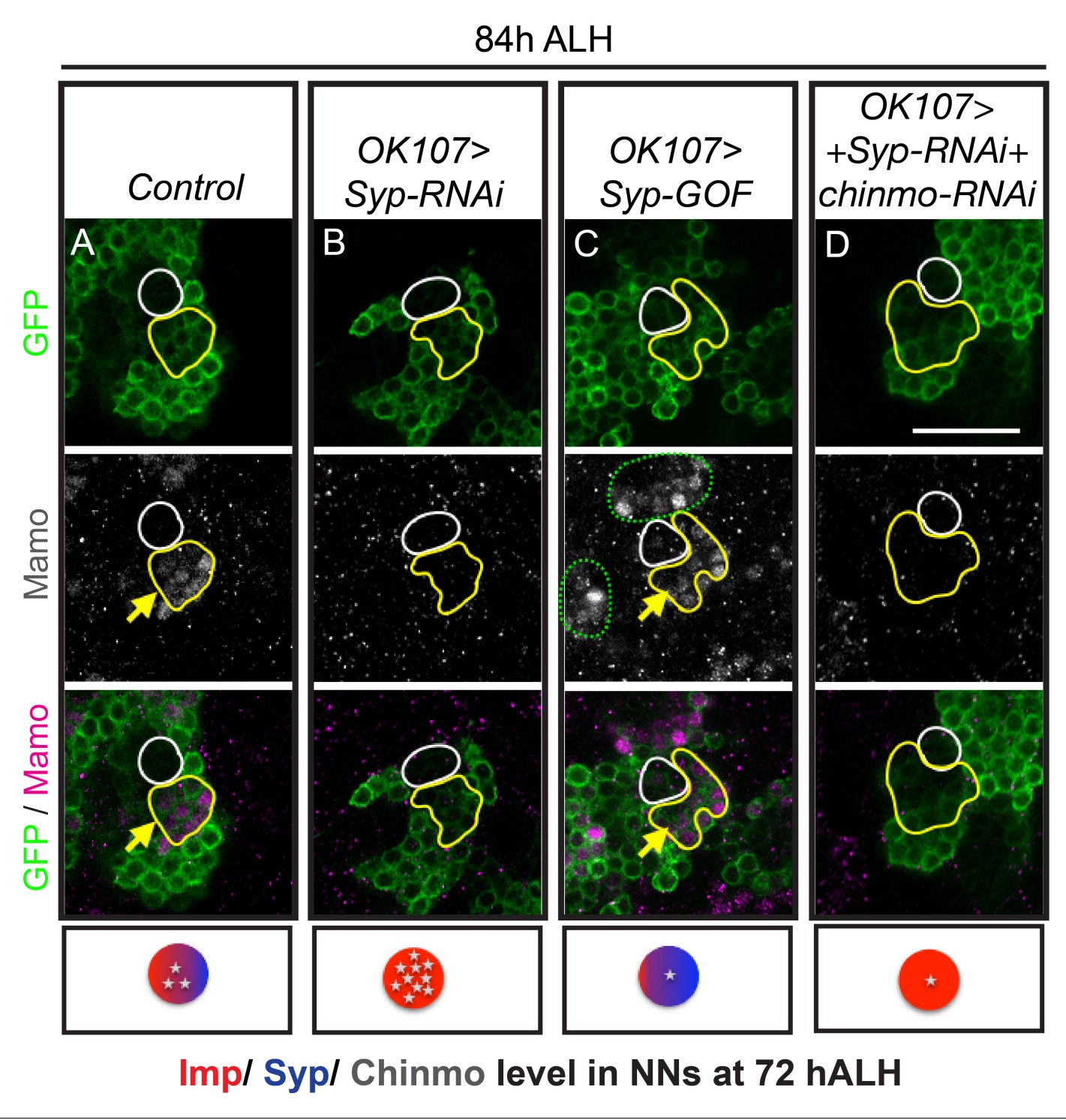

**Figure 3.** Mamo protein expression requires Syp RNA binding protein. MB lineages (*OK107 > GFP*) immunostained for GFP and Mamo. A single focal plane near the MB NB is shown. Newborn neurons (NN) are outlined in white and young/maturing neurons are outlined in yellow. Arrows indicate regions with Mamo protein expression. Images are representative of n > 18. Scale bar = 20 μm. The diagram below shows the relative protein levels of Imp (red), Syp (blue), and Chinmo (stars) expressed in the young/maturing neurons when they were newborn 12 hr prior (based on *Figure 3—figure supplement 1* and *Liu et al., 2015*). (A-D) Mamo expression in young/maturing neurons occurs in genotypes with low Chinmo levels (A and C) with the exception of *Syp-RNAi* plus *chinmo-RNAi* (D). Note that green dashed circle is labeling other MB neurons. Chinmo immunostaining and quantifications from earlier stages can be found in *Figure 3—figure supplement 1*. Mamo levels are shown in *Figure 3—figure supplement 2*.

DOI: https://doi.org/10.7554/eLife.48056.009

*Figure 3 continued on next page*

*Figure 3 continued*

The following source data and figure supplements are available for figure 3:

**Source data 1.** Mamo staining intensity in young/maturing neurons with Syp manipulations.

DOI: https://doi.org/10.7554/eLife.48056.012

**Source data 2.** Chinmo staining intensity in newborn neurons with different genetic manipulations.

DOI: https://doi.org/10.7554/eLife.48056.013

**Figure supplement 1.** Syp gradient alter Chinmo protein expression.

DOI: https://doi.org/10.7554/eLife.48056.010

**Figure supplement 2.** Mamo is absent with premature low Chinmo levels before 60 hr ALH.

DOI: https://doi.org/10.7554/eLife.48056.011

## Mamo is necessary for the middle α'/β' fate

Given Chinmo's role in specifying both γ and α'/β' neurons and in Mamo's expression in the middle, α'/β' temporal window, we hypothesize that Mamo is crucial for the specification of the α'/β' neuronal fate. To more accurately distinguish different neuronal types, we used a combination of FasII/Trio staining to examine MB neuronal projections and cell body markers (Abrupt and Trio) (*Figure 5A*). In keeping with our hypothesis, *mamo-RNAi* caused the α'/β' lobes to essentially vanish (*Figure 5C*). Wildtype α'/β' cell bodies characteristically express strong Trio, both in the cytoplasm and plasma membrane (*Awasaki et al., 2000*) (*Figure 5B and B'*), whereas we detected little to no Trio expression in MB neurons expressing *mamo-RNAi* (*Figure 5C'*). This demonstrates a significant role for Mamo in proper α'/β' fate specification.

Nevertheless, Mamo may be pleiotropic, as the number of cells expressing the γ-specific marker, Abrupt, is also reduced (*Figure 5C'*). Examination of larval MB markers suggest that γ specification is normal (data not shown), signifying a distinct role for Mamo in γ neuron biology, which is being investigated separately. As γ lobe phenotypes complicate analysis of Mamo's role in α'/β' fate specification, we examined the role of Mamo in MBs after eliminating a majority of γ neurons with *chinmo-RNAi*. With premature weak Chinmo (*chinmo-RNAi*), γ neuron production significantly (p<0.001) decreased from $38 \pm 0.3\%$ in control animals to $18 \pm 1.6\%$ with *chinmo-RNAi* (*Figure 5D and D'* and *Figure 5—figure supplement 3*). This suggests an early onset of α'/β' neuron production and is consistent with the early Mamo transcription initiation at 48 hr ALH (*Figure 4F* and *Figure 4—figure supplement 1*) and early Mamo expression at 60 hr ALH (*Figure 2F*) that we detected with *chinmo-RNAi*. However, the percentage of α'/β' cells did not significantly increase. This suggests that with *chinmo-RNAi*, the window of α'/β' production was shifted earlier, rather than prolonged. The result is an increase in α/β neurons (*Figure 5D and D'* and *Figure 5—figure supplement 3*). The percentage of α/β-characteristic Trio⁻/Abrupt⁻ cells increased from $41 \pm 0.5\%$ in control to $64 \pm 1.1\%$ with *chinmo-RNAi* (p<0.001). Strikingly, the combination of *mamo-RNAi* and *chinmo-RNAi* completely eliminated α'/β' neuronal features (*Figure 5E and E'*). This result substantiates an essential role for Mamo in α'/β' temporal fate determination.

## Mamo variant functioning in α'/β' fate specification

Next, we set out to determine which Mamo protein variant acts in α'/β' fate specification. The *mamo* gene has seven splice isoforms that produce four protein variants (*Figure 5—figure supplement 1A and B*). One variant contains only a BTB domain, and the remaining three have a BTB domain and three to five zinc finger motifs. We tested the three variants containing zinc fingers (codon optimized and lacking UTRs and introns) and found that only the Mamo variant with four zinc fingers (4ZF: corresponding to *mamo* isoforms RG and RF) produced MBs that were not reduced in size, but had reduced Fas-II labeling (*Figure 5—figure supplement 1C–F*), reminiscent of a shift in temporal fate specification. Importantly, we confirmed that the 4ZF Mamo construct could override the *mamo*-RNAi phenotype (*Figure 5—figure supplement 1G and H*). These data strongly indicate that either the *mamo-RF* or *mamo-RG* isoform lies downstream of Imp/Syp gradients and Chinmo in α'/β' temporal fate determination. In further support of a hierarchy, 4ZF Mamo overexpression does not alter Imp, Syp or Chinmo levels (*Figure 5—figure supplement 2*). The remainder of analyses in this paper were performed with the 4ZF Mamo transgene.

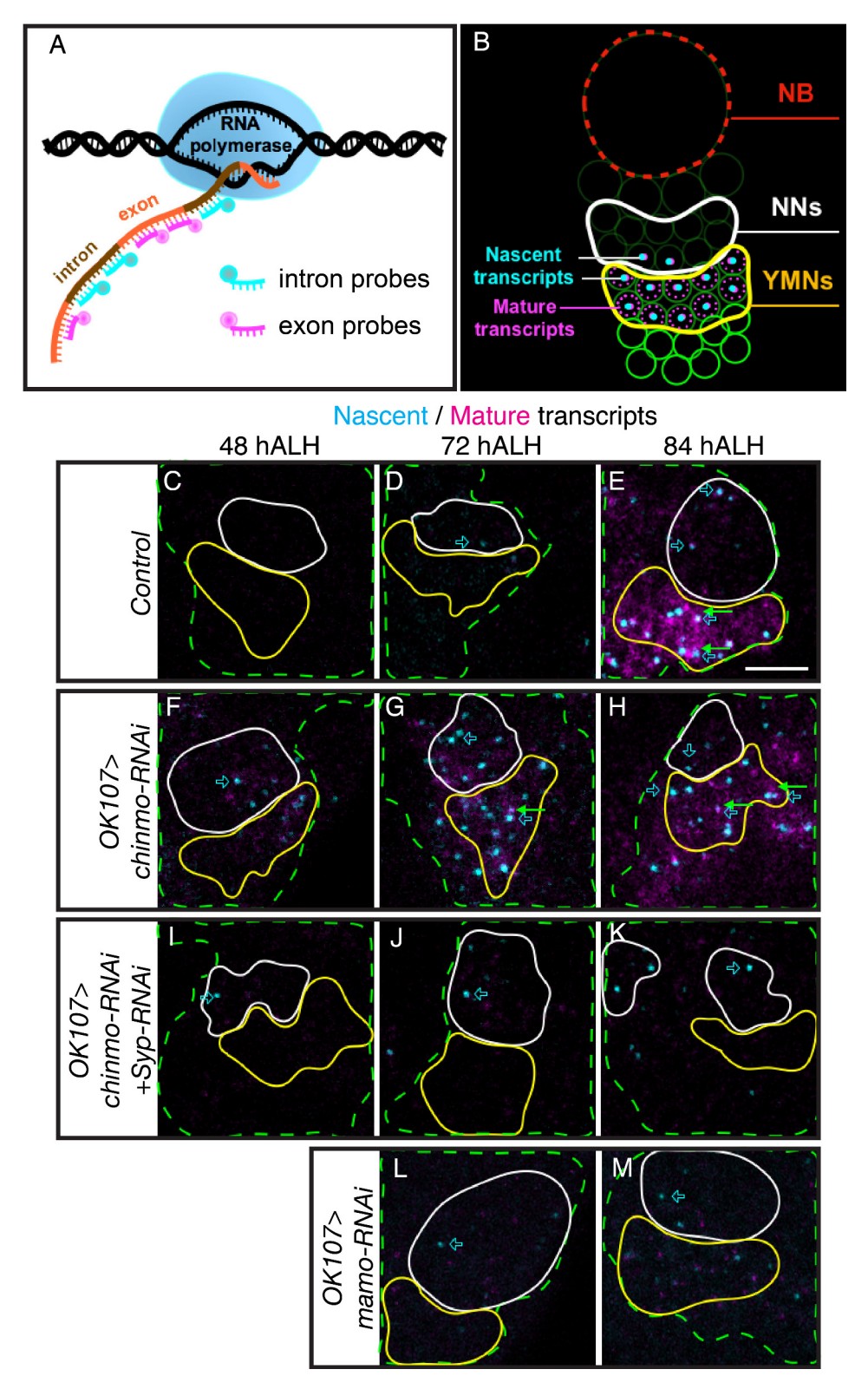

**Figure 4.** Syp promotes sustained *mamo* transcription. (A) Graphic illustrating the use of intron and exon probes for single molecule florescent in situ hybridization (smFISH). Nascent transcripts are labeled by both intron and exon probes, while mature transcripts are only labeled by exon probes. (B) Diagram illustrating interpretation of smFISH data. Active transcription is seen as a single, double-labeled focus per cell. Mature transcripts (magenta only) are diffuse and cytoplasmic. (C–M) smFISH with probes targeting *mamo* intronic (cyan) or exonic (magenta) sequences. Images are of developing

*Figure 4 continued on next page*

eLIFE Research article

*Figure 4 continued*

larval brains with different genetic manipulations of the MB. Maximum Intensity Z-projections (2.3–3.8 μm) near the MB NB are shown. MB cells are determined by OK107 >GFP (green dashed outline) and the newborn neuron (NN) region is outlined in white and the young/maturing neuron (YMN) region is outlined in yellow. Blue open arrows highlight examples of *mamo* active transcription, green arrows highlight examples of mature *mamo* transcripts. Images are representative of n > 6. Scale bar = 5 μm. Control brains (OK107 >GFP) show active transcription in NNs at 72 hr (**D**) and in NNs and young/maturing neurons at 84 hr (**E**). Mature transcripts are visible in young/maturing neurons at 84 hr ALH (**E**). OK107 >chinmo-RNAi results in a shift in the timing of *mamo* transcription. Active transcription is visible at 48 hr in both NNs and young/maturing neurons (**F**) and is abundant at 72 hr and 84 hr ALH (**G and H**). Note that MBs expressing *chinmo-RNAi* together with *Syp-RNAi* have active transcription in NNs at all time points, but lack mature transcripts and active transcription in young/maturing neurons (**I–K**). Depleting *mamo* (OK107 >mamo-RNAi) causes loss of mature transcripts and active transcription in young neurons (**L–M**). The quantification of *mamo* mature transcripts is in *Figure 4—figure supplement 1*.
DOI: https://doi.org/10.7554/eLife.48056.014

The following source data and figure supplement are available for figure 4:

**Source data 1.** Quantifications of mature *mamo* transcript.
DOI: https://doi.org/10.7554/eLife.48056.016
**Source data 2.** Intron and exon probe sequences.
DOI: https://doi.org/10.7554/eLife.48056.017
**Figure supplement 1.** The intensity quantification of *mamo* mature transcripts in different manipulations of MB neurons.
DOI: https://doi.org/10.7554/eLife.48056.015

## Mamo is sufficient to promote the middle α′/β′ fate

We next set out to determine whether Mamo could drive α′/β′ neuronal fate outside of the middle developmental window. Continuous expression of transgenic Mamo greatly enlarged the α′/β′ lobes and drastically reduced the thickness of the α/β lobes (*Figure 5F*), as determined by Trio and Fas-II expression. The reciprocal changes in Trio and Fas-II expression indicate an extension of α′/β′ production into the pupal stage, when wildtype MB NBs produce α/β neurons. The alterations in the MB lobes are consistent with that of the cell body region which had enhanced and expanded Trio$^{PM,Cyto}$ expression (*Figure 5F′* and *Figure 5—figure supplement 3*), denoting increased numbers of α′/β′ neurons. The percentage of α′/β′-characteristic Trio$^{PM,Cyto}$ expressing cells increased from 20 ± 0.8% in control to 32 ± 1.8% with *mamo-GOF* (p<0.005). Moreover, there was a concomitant reduction in Trio/Abrupt double negative α/β domains (*Figure 5F′* and *Figure 5—figure supplement 3*). The percentage of α/β-characteristic Trio$^-$ expressing cells decreased from 41 ± 0.5% in control to 2 ± 0.4% with *mamo-GOF* (p<0.001). Despite the lack of a clearly identifiable γ lobe, many of the cell bodies were Abrupt/Trio double positive (*Figure 5F′*, red arrow), leaving reservations about Mamo's ability to transform the early-born γ to α′/β′ fate.

To definitively determine whether Mamo can transform γ neurons to the α′/β′ fate, we overexpressed Mamo in MBs that would otherwise produce only γ neurons. We accomplished this using *Syp-RNAi* with which NBs do not appear to age (*Yang et al., 2017*). As previously reported (*Yang et al., 2017*), the *Syp-RNAi* expressing NBs cycled incessantly (*Figure 5G*, NBs marked with asterisks). Moreover, temporal progression stalled, producing only γ neurons, as determined by both lobe morphology and marker expression (*Figure 5G and G′*). The addition of transgenic Mamo caused the majority of neurons to assume the α′/β′ fate (*Figure 5H and H′*). Importantly, this Mamo transgene does not contain UTRs or introns, thus it is not subject to post-transcriptional regulation. Note that Mamo overexpression did not alter the continued NB division (*Figure 5H′*, asterisks). These results confirm that Mamo is both necessary and sufficient to determine the middle α′/β′ temporal fate. Taken together, our findings indicate that Mamo acts as a key temporal fate determinant for the α′/β′ neuronal fate in the serial temporal fate diversification of MB neurons.

## Mamo maintains α′/β′ cell-fate

As Mamo is essential for α′/β′ neuronal fate specification and continues to be expressed in adult α′/β′ neurons (*Figure 6—figure supplement 1C*), we wanted to examine whether it is required to maintain α′/β′ cell fate. We utilized a temperature sensitive GAL80 (*McGuire et al., 2003*) to limit the expression of *mamo-RNAi* until after neuron fate was established. A temperature shift from 18°C to 29°C induced the expression of *mamo-RNAi* (*Figure 6A*). After adult eclosion, expression of *mamo-RNAi* for 21 days was required to effectively reduce Mamo protein levels (*Figure 6—figure supplement 1*). When Mamo was efficiently knocked down in adult MBs, the α′/β′-characteristic

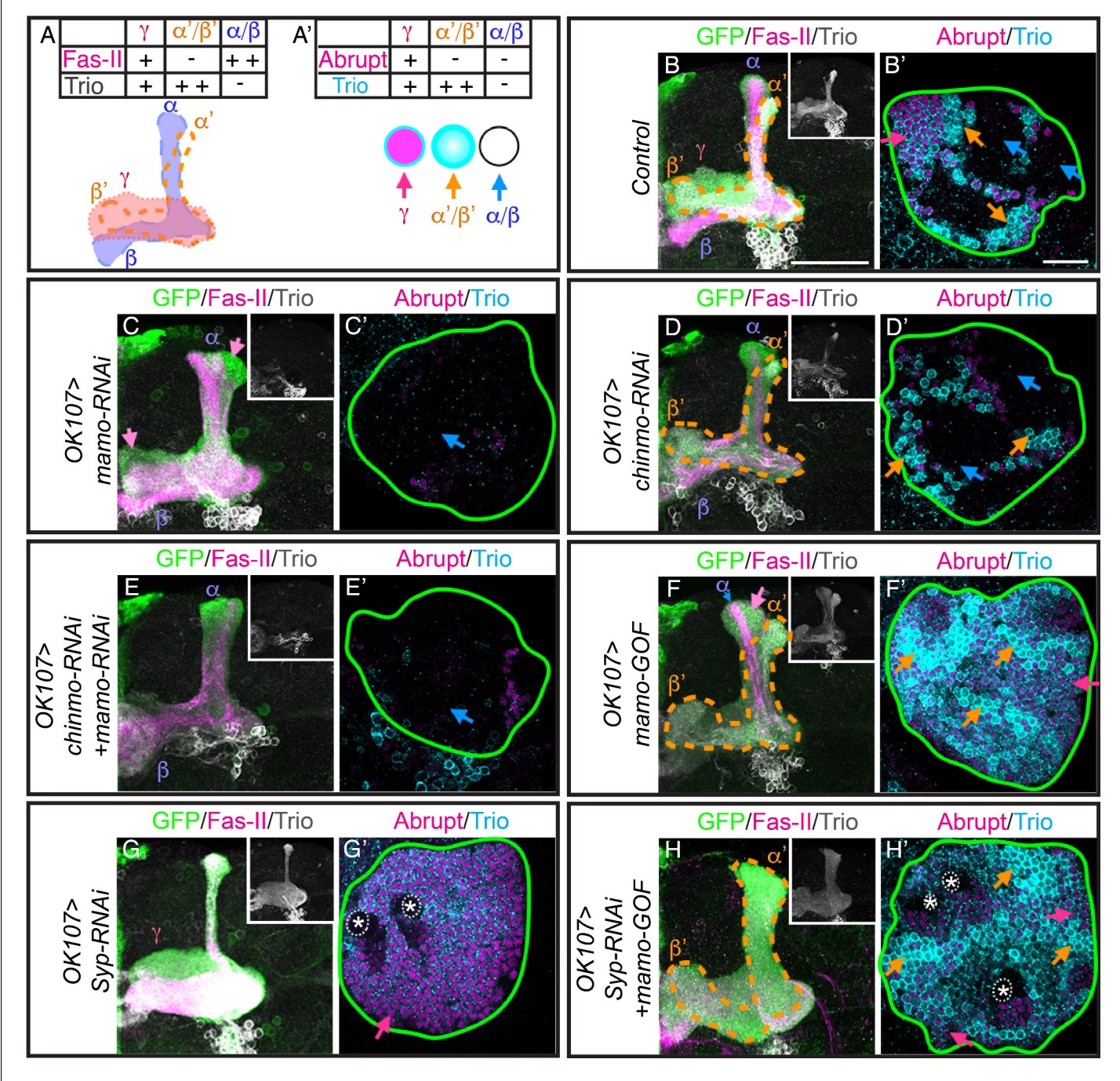

**Figure 5.** Mamo is necessary and sufficient for the α′/β′ fate. (**A**) Schematic of MB lobe morphology and table of corresponding marker expression. (**B–H**) Adult MB lobes (*OK107 > GFP*) immunostained for GFP, Fas-II and Trio. Images are Z-stack projections of the axon region and are representative of n > 18. Lobes are identified based on both 3D structure and marker expression. α′/β′ lobes are outlined with orange dashed lines. Scale bar = 50 μm. Insert shows Trio staining alone. (**A′**) MB cell body markers that distinguish three MB neuron types. (**B′–H′**) Adult MB cell bodies (*OK107 > GFP*) immunostained for Abrupt, Trio and GFP. GFP channel is not shown, but is represented by a green outline. Colored arrows highlight MB neuron types red=γ ( Trio$^{PM}$/Abrupt$^+$ ), orange=α′/β′ (Trio$^{PM,Cyto}$/Abrupt$^-$), blue=α/β (Trio$^-$/Abrupt$^-$). Images are representative of n > 6. A single focal plane is shown. Scale bar = 20 μm. Note wide Fas-II$^{++}$ α/β lobes and a morphologically indistinct FasII weak/negative lobe (magenta arrows) with *mamo-RNAi* (**C**). *mamo-RNAi* and *chinmo-RNAi + mamo-RNAi* both lack most cell body marker staining (**C′ and E′**). γ lobes and cell body markers are reduced in *chinmo-RNAi* alone (**D and D′**). Mamo overexpression (*mamo-GOF*) results in an expanded α′/β′ lobe (**F**) and increased numbers of Trio$^{PM,Cyto}$/Abrupt$^-$ cell bodies (**F′′**). Note the Fas-II$^{++}$, Trio$^-$ axons in the A/P (α), but not medial/lateral (β) portion of the axon lobe (blue arrow) which is surrounded by FasII$^{-/weak}$, Trio$^+$, morphologically indistinct axons (magenta arrow) in *mamo-GOF* (**F**). The cell body region is overwhelmingly Trio$^+$ (**F′**). *Syp-RNAi* MB

*Figure 5 continued on next page*

*Figure 5 continued*

(G and G') shows only γ neurons (note the A/P axon bundle characteristic of un-remodeled γ neurons). *Syp-RNAi* plus *mamo-GOF* produced expanded α'/β' lobes (H) and mostly Trio^PM,Cyto/Abrupt^- cell bodies, with some Trio^PM/Abrupt^+ cells (H'). Note proliferating NBs (*) and adjoining unspecified young/maturing neurons produced with *mamo-GOF* (G' and H'). The analysis of Mamo variants is in *Figure 5—figure supplement 1*. The analysis of hierarchical model is in *Figure 5—figure supplement 2*. The quantification of neuron populations is in *Figure 5—figure supplement 3*.

DOI: https://doi.org/10.7554/eLife.48056.018

The following source data and figure supplements are available for figure 5:

**Source data 1.** Quantification of adult MB neuron types.

DOI: https://doi.org/10.7554/eLife.48056.022

**Source data 2.** Intensity of Imp/Syp/Chinmo staining.

DOI: https://doi.org/10.7554/eLife.48056.023

**Figure supplement 1.** The Mamo variant containing 4ZFs is the prospective isoform acting in α'/β' temporal fate determination.

DOI: https://doi.org/10.7554/eLife.48056.019

**Figure supplement 2.** Mamo acts as a downstream factor of Imp/Syp/Chinmo gradients.

DOI: https://doi.org/10.7554/eLife.48056.020

**Figure supplement 3.** The quantification of MB neuron types.

DOI: https://doi.org/10.7554/eLife.48056.021

cytoplasmic Trio (Trio$^{Cyto,PM}$/Abrupt$^-$) was completely absent (*Figure 6C and C'*, 22 ± 1.6% control vs 0 ± 0% *mamo-RNAi*, p<0.0005). The remaining membrane Trio expression (Tio$^{PM}$/Abrupt$^-$, 17 ± 2% of cells) may require a longer course of *mamo-RNAi* to be eliminated. These results clearly demonstrate that Mamo is required to maintain α'/β' cell-type specific Trio expression.

## Mamo stimulates α'/β' specific gene expression in mature MB neurons

To examine whether Mamo is sufficient to promote α'/β' fate transformation in mature MB neurons, we performed a similar experiment—overexpressing Mamo after neuron fate is established. We induced the expression of the 4ZF Mamo transgene by deactivating GAL80 with a temperature shift (*Figure 6A*). Overexpressing Mamo in adult MB neurons resulted in a modest, but significant increase in the percentage of cells with α'/β' characteristic gene expression (Trio$^{PM,Cyto}$/Abrupt$^-$, 22 ± 1.6% control vs 42 ± 4.1% *Mamo-GOF*, p=0.01; *Figure 6D and D'*). Markedly, the ability of Mamo overexpression to transform MB neurons diminished over time, as transgene induction at pupal development was more effective at increasing Trio$^{PM,Cyto}$/Abrupt$^-$ cells (*Figure 6—figure supplement 2*). Taken together, our data suggests that Mamo acts as a terminal selector transcription factor for α'/β' neuronal fate, in part by regulating Trio gene expression.

## Mamo is regulated by weak Chinmo and Syp in antennal lobe development

While the MB is a well-characterized lineage, with only three main temporal fates and constant NB division from embryonic through pupal stages, it is not necessarily typical. We therefore wanted to examine whether Mamo was downstream of Imp, Syp and Chinmo in other lineages. The AL lineages produce many more temporal fates over a shorter period of time (one AL NB produces 22 postembryonic fates). Interestingly, Imp and Syp temporal protein gradients show distinct lineage-characteristic expression levels and rates of gradient progression (*Liu et al., 2015*; *Ren et al., 2017*; *Syed et al., 2017*). MB NBs have shallow, slowly progressing gradients and AL NBs have steep, rapidly progressing gradients (*Liu et al., 2015*). We therefore examined Mamo in AL lineages defined by GR44F03-KD (*Awasaki et al., 2014*). We found Mamo expression in at least two of the four labeled AL lineages at 84 hr ALH (*Figure 7D*). The defined temporal expression window leads us to believe that Mamo regulation by weak Chinmo and Syp may serve as a general mechanism for specifying temporal fate windows. To corroborate this idea, we monitored Chinmo and Mamo expression in AL lineages. We compared wildtype ALs with ALs expressing either *chinmo-RNAi* or both *chinmo-RNAi* and *Syp-RNAi*. Similar to our findings in the MB, weak Chinmo (*chinmo-RNAi*) induced precocious Mamo expression (*Figure 7B*) and an increase in the number of Mamo-positive neurons at 84 hr ALH (*Figure 7F*). Moreover, Mamo expression was lost when Syp was repressed (*Figure 7C and G*). These findings in the AL combined with our previous MB data leads us to a model where weak

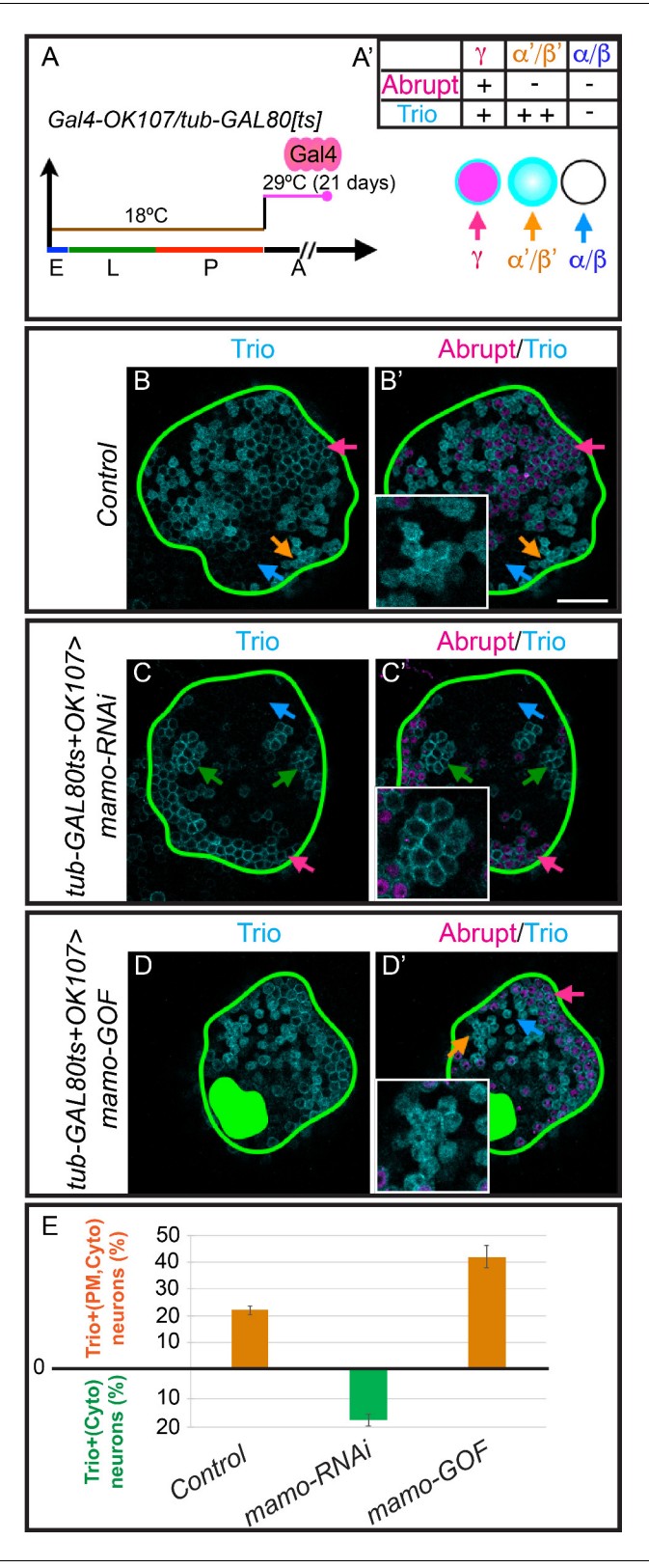

**Figure 6.** Mamo maintains α'/β' MB neuron markers. (**A**) Scheme for temperature shift assay. Temperature-sensitive GAL80 was inactivated for 21 days after adult eclosion. E = embryo, L = larva, p=pupa, A = adult (**A'**) MB cell body marker expression of the three MB neuron types. (**B–D**) Adult MB cell bodies (*OK107 > GFP+tub-GAL80ts*) immunostained for GFP, Abrupt, and Trio. Colored arrows identify MB neuron types based on marker expression: red=γ (Trio^PM/Abrupt+), orange=α'/β' (Trio^PM,Cyto /Abrupt-), blue=α/β (Trio-/Abrupt-). Images are representative of n > 6. A single focal plane is shown.
*Figure 6 continued on next page*

*Figure 6 continued*

Scale bar = 20 µm. Notice loss of cytoplasmic Trio staining after depleting expressing *mamo-RNAi* (**C and C'**, green arrows = Trio[PM]/Abrupt⁻ neurons). Overexpressing Mamo (*mamo-GOF)* after adult eclosion results in increased numbers of Trio[PM,Cyto]/Abrupt⁻ neurons (green filled area indicates MB calyx) (**D and D'**). (**E**) Percent of MB neurons expressing both cytoplasmic and plasma membrane Trio in orange (Trio[PM,Cyto]/Abrupt⁻) and percent of neurons expressing only plasma membrane Trio (Trio[PM]/Abrupt⁻) in green  (mean ± SEM, n = 3 brains). Time course of Mamo depletion with adult induced RNAi is shown in *Figure 6—figure supplement 1*.

DOI: https://doi.org/10.7554/eLife.48056.024

The following source data and figure supplements are available for figure 6:

**Source data 1.** Quantification of Trio expression (PM, Cyto).
DOI: https://doi.org/10.7554/eLife.48056.027
**Figure supplement 1.** Temperature shift assay for effectively repressing Mamo.
DOI: https://doi.org/10.7554/eLife.48056.025
**Figure supplement 2.** Mamo stimulates α'/β' specific gene expression.
DOI: https://doi.org/10.7554/eLife.48056.026

Chinmo and Syp specifically guide Mamo expression in defined temporal windows of diverse lineages (*Figure 8*).

## Discussion

### Weak Chinmo on Mamo protein expression

Chinmo levels in newborn neurons correlate with adult neuron identity (*Kao et al., 2012*; *Zhu et al., 2006*). Based on smFISH, *mamo* transcription is initiated in newborn MB neurons around 72 hr ALH (*Figure 4D*), which corresponds to weak Chinmo expression (*Figure 3—figure supplement 1G*). Moreover, Mamo is only expressed when Chinmo levels are low, as Mamo is not expressed after either eliminating or overexpressing Chinmo (*Figure 2*). Together these data indicate that low Chinmo levels activate *mamo* transcription in young/maturing neurons.

Transcription initiation is not the only requirement for Mamo protein expression; Syp is also required (*Figures 3* and *4*, discussed below). This could explain why we do not see Mamo expression turn on in γ neurons (*Figure 1—figure supplement 1B*), even as they age and Chinmo levels decrease, becoming quite low around wandering larval stage (*Zhu et al., 2006*). γ neurons begin to express Mamo later, around pupation (*Figure 1—figure supplement 1C*), despite lacking Syp (*Liu et al., 2015*). It has not yet been tested whether weak Chinmo levels are required for later Mamo expression in γ neurons. It is therefore possible that Mamo expression is controlled at this stage by an additional factor(s).

### Chinmo, a potential morphogen

ChIP-chip performed in embryos found five Chinmo binding sites within the *mamo* gene (*Roy et al., 2010*), consistent with direct activation of *mamo* transcription. However, the nature of Chinmo's concentration dependent actions is still unclear. Some morphogens such as Bicoid bind different targets at increasing concentrations based on the affinity of binding to different sites as well as the chromatin accessibility of the binding sites (*Hannon et al., 2017*). This may also be the case with Chinmo, but would not easily explain why Mamo expression is inhibited at higher Chinmo concentrations. The gap gene Krüpple, on the other hand, has concentration dependent activities at the same binding site. Krüpple acts as an activator at lower concentrations and as a repressor at high concentrations (*Sauer and Jäckle, 1991*). Krüpple's C-terminus has the ability to activate genes and is also the location for dimerization. Upon dimerization, the C-terminus can no longer activate genes and Krüpple transforms from an activator to a repressor (*Sauer and Jäckle, 1993*). Our data suggests that low concentrations of Chinmo activate *mamo*. However, in the testis, Chinmo is suspected to function as a transcriptional repressor (*Flaherty et al., 2010*; *Grmai et al., 2018*). It is feasible that Chinmo, like Krüpple, could switch from an activator to a repressor. The protein concentration would affect whether Chinmo is a monomer (in the presence of other BTB proteins, a heterodimer) or a homodimer, and thus potentially which cofactors are recruited.

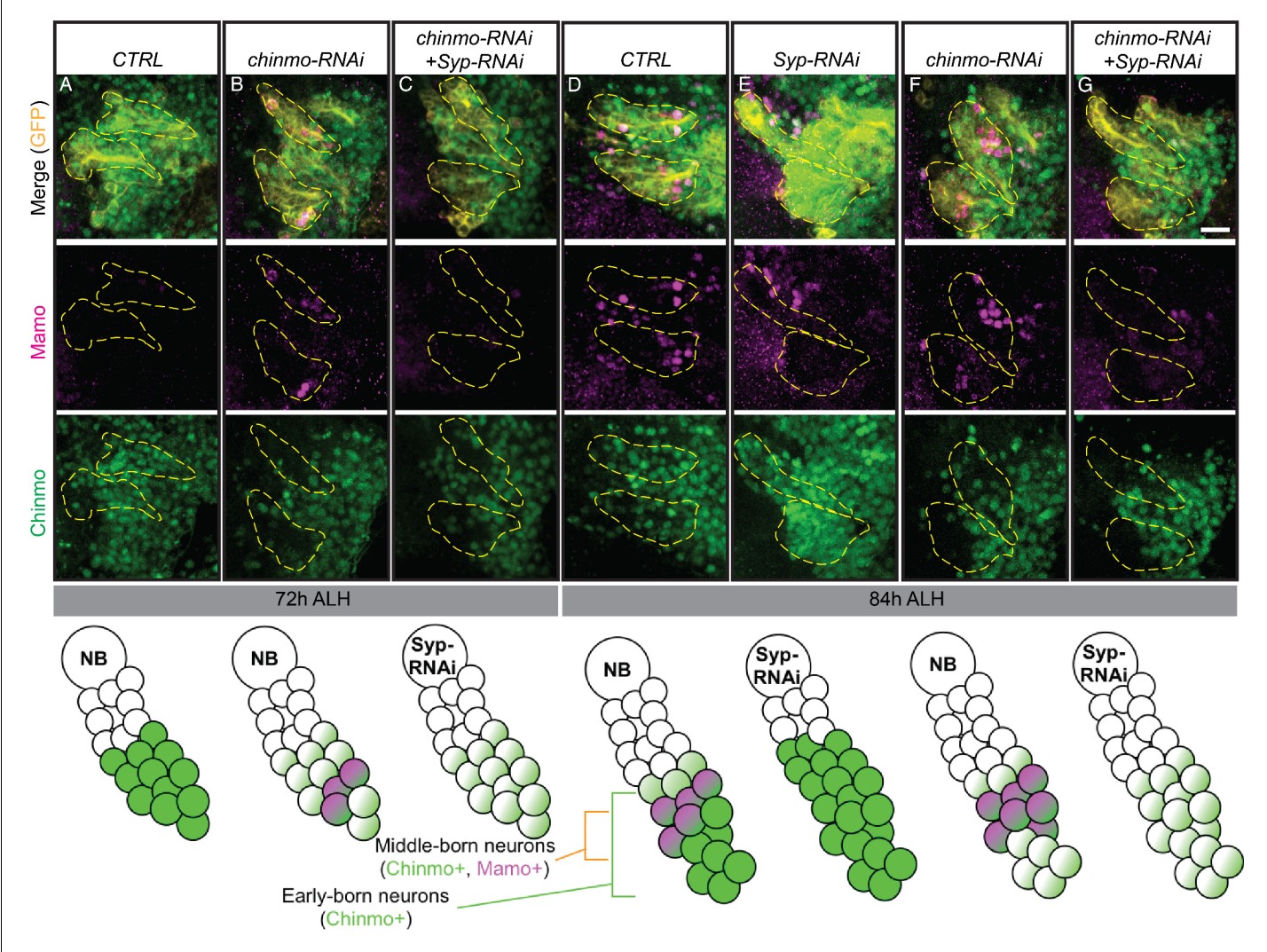

**Figure 7.** Weak Chinmo drives Mamo expression in AL lineages. Immunostaining for GFP (yellow), Mamo (magenta) and Chinmo (green) in AL lineages expressing *chinmo-RNAi*, *Syp-RNAi*, or dual *chinmo/Syp-RNAi*. Images are Z-stack projections (standard deviation) of the cell body region and are representative of n > 10. Two AL lineages are outlined with yellow dashed lines based on GR44F03 lineage restricted actin >GFP expression. GR44F03 lineage restricted actin-LexA also drives RNAi transgenes. Scale bar = 10 μm. The diagrams below summarize the protein levels of Mamo and Chinmo. (A-C) 72 hr ALH larval brains. Control brains have no visible Mamo staining within the AL lineages (A). Mamo staining is visible after reducing Chinmo levels with *chinmo-RNAi* (B). *chinmo-RNAi+Syp-RNAi* results in reduced Chinmo, but no Mamo expression (C). (D-G) 84 hr ALH. Control brains have Mamo positive cells in AL lineages (D). *Syp-RNAi* produces expanded Chinmo and loss of Mamo expression (E). *chinmo-RNAi* reduces Chinmo and increases the number of Mamo positive cells (F). *chinmo-RNAi + Syp-RNAi* results in weak Chinmo expression and loss of Mamo staining (G).
DOI: https://doi.org/10.7554/eLife.48056.028

## Syp stabilizes *mamo* transcripts

The ascending Syp RNA binding protein temporal gradient regulates Mamo expression both indirectly via its inhibition of Chinmo and also presumably directly, interacting with the *mamo* transcript and promoting its expression. The bi-modal, transcriptional (Chinmo) and post-transcriptional (Syp), regulation of the Mamo terminal selector is extremely advantageous. Given our finding that Mamo expression is positively autoregulated (*Figure 4L and M*) and that Mamo continues to be expressed into adult neurons (*Figure 6—figure supplement 1C*), it is particularly important to control the timing of Mamo's onset. The additional layer of post-transcriptional regulation adds an extra safeguard, helping to guarantee that neuronal temporal patterning is a robust system. Indeed, as brain

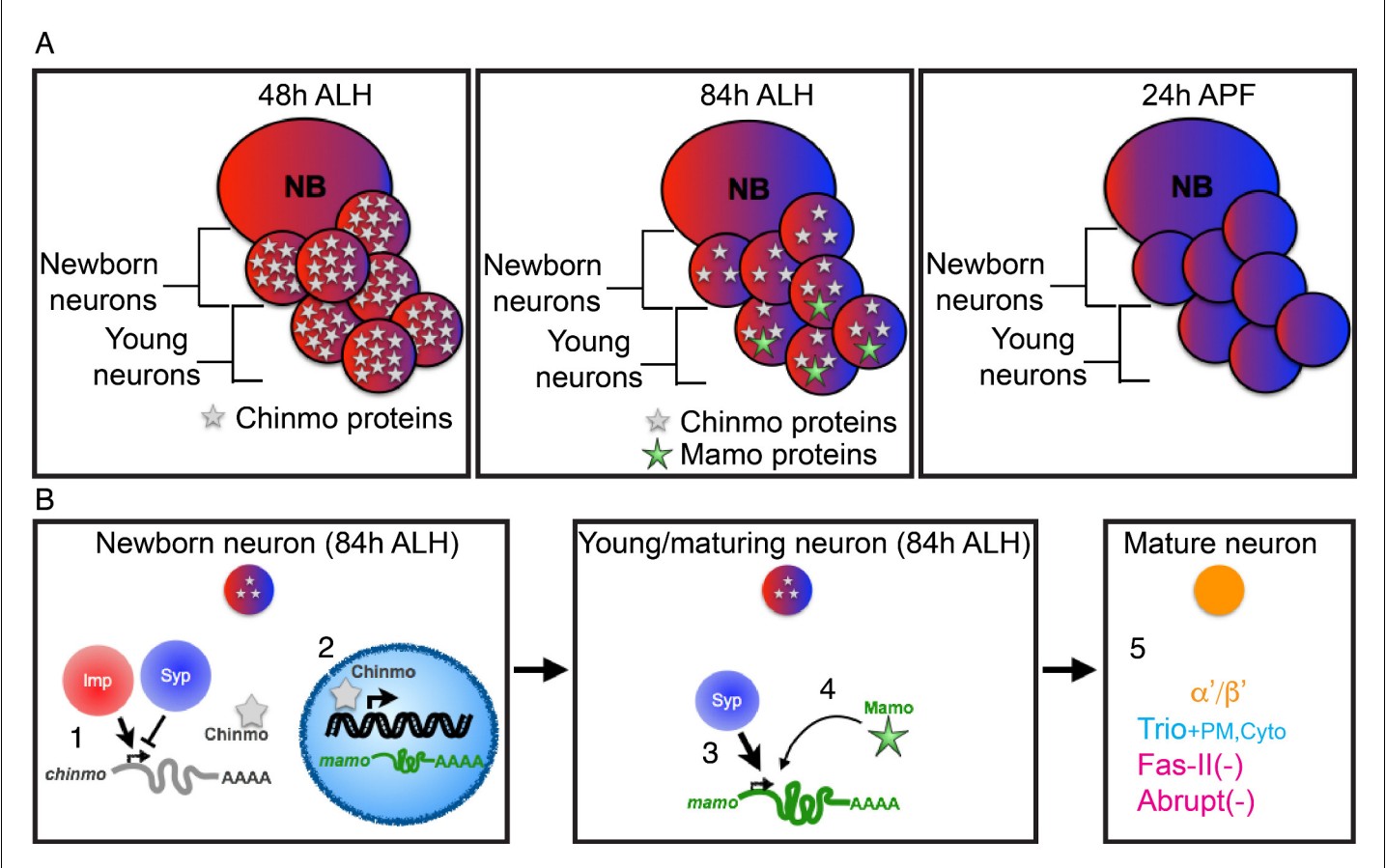

**Figure 8.** Schematic of α'/β' neuronal fate determination. (**A**) Diagram of the three temporal windows of MB development. Images are color coded to illustrate the expression level of Imp (red) and Syp (blue). Chinmo (gray stars) and Mamo (green stars) levels are also indicated. (**B**) Hierarchical regulation of α'/β' neuronal fate determination. (1) Balance of Imp and Syp affects *chinmo* translation in the newborn neuron, producing low Chinmo levels (*Liu et al., 2015*). (2) Low Chinmo levels initiate *mamo* transcription in the newborn neuron. (3) Syp promotes *mamo* mRNA maturation/ stabilization during neuron maturation. (4) Mamo positively autoregulates its own expression. (5) Mamo promotes α'/β' specific gene expression in the mature neuron.

DOI: https://doi.org/10.7554/eLife.48056.029

development needs to adapt to environmental conditions such as nutrient deprivation, it is crucial to ensure that there is no loss of neuronal diversity (*Lanet and Maurange, 2014*; *Lin et al., 2013*).

Syp is a homolog of mammalian SYNCRIP (synaptotagmin-binding cytoplasmic RNA-interacting protein) also known as hnRNP-Q. SYNCRIP is involved with multiple facets of mRNA regulation including mRNA splicing and maturation (*Mourelatos et al., 2001*), mRNA localization and stabilization (*Bannai et al., 2004*) as well as inhibiting mRNA translation and miRNA-mediated repression via competition with Poly(A) binding proteins (*Svitkin et al., 2013*). The *Drosophila* ortholog seems to have corresponding functions. *Drosophila* Syp was isolated from the spliceosome B complex, indicating a conserved role in mRNA splicing (*Herold et al., 2009*). Syp has likewise been found to operate in mRNA localization and stabilization (*McDermott et al., 2012*). Furthermore, it has clear roles in altering protein expression of its mRNA targets, both positively and negatively (*McDermott et al., 2014*). The bidirectional influence on protein expression likely reflects different Syp modalities.

In this study, we show that Syp is required for Mamo protein expression in the MB and AL neuronal lineages (*Figures 3* and *7*). To determine the nature of this regulation, we performed smFISH (*Figure 4*). In the absence of Syp, *mamo* transcription was initiated prematurely in response to weak Chinmo levels, yet mature transcripts failed to accumulate (*Figure 4L–K*). This leads us to believe that Syp directly binds *mamo* mRNA and aids in its splicing, maturation and/or stabilization. This is

consistent with our finding that overexpressing a *mamo* cDNA (lacking 5' UTR, 3' UTR and introns) was able to promote cell fate changes despite repression of Syp (*Figure 5H*).

## Mamo is necessary and sufficient for α'/β' temporal fate

Mamo is required to produce the α'/β' neurons in the middle temporal window of the MB lineages. Trio positive α'/β' neurons are clearly absent after RNAi depletion of Mamo during development (*Figure 5C,C' and E,E'*). Cell production does not appear to be altered, as *mamo-RNAi* expressing MBs are a normal size. This begs the question of which, if any terminal fate the middle-born neurons adopt in the absence of Mamo. The limited markers for each MB cell type makes it difficult to determine whether the middle-born neurons undergo fate transformation or simply lack terminal fate. The presence of a Fas-II negative lobe (*Figure 5C*, magenta arrow) hints that some middle-born neurons may not carry temporal fate information, but phenotypic analysis is complicated by defects in γ neuron maturation/remodeling. Removing the γ neurons with *chinmo-RNAi* eliminates this complication, but it is still unclear whether, without Mamo, the neurons are transformed to the α/β fate (*Figure 5E and E'*). The Fas-II positive, α/β lobe appears enlarged (*Figure 5E*), but it is difficult to tell whether all axons are Fas-II positive or whether Fas-II negative axons are comingled with α/β axons. Without a cell type-specific, cell body marker for α/β neurons, it is ambiguous whether the middle-born cells are transformed to α/β or whether they simply lack α'/β' temporal fate. A transformation to α/β fate would suggest that either α/β is the default fate of MB neurons (requiring no additional terminal selector) or that Mamo expression inhibits α/β specific factors.

Mamo's role in promoting α'/β' fate is further supported by Mamo overexpression phenotypes. Overexpression of Mamo in the MB is able to transform α/β and γ neurons to α'/β' neurons (*Figure 5F,F', H and H'*). In an otherwise wildtype scenario, overexpression of *mamo* did not transform every cell to α'/β' fate (*Figure 5F*). Instead the α'/β' lobe was expanded and the other lobe seemed to be an amalgam of α and γ like lobes. This could be due to incomplete penetrance/low expression levels of the *mamo* transgene or it is possible that the α/β and γ cells retain their own terminal selector driven, cell-type specific gene expression, thus complicating the fate of the differentiated neuron. Mamo overexpression does not alter the specification factors Imp, Syp or Chinmo (*Figure 5—figure supplement 2*) and presumably there are terminal selector genes expressed downstream of high Chinmo and possibly in Chinmo-absent cells. This seems a likely possibility when overexpressing Mamo in γ neurons. With *Syp-RNAi*, NBs are 'forever young' and divide into adulthood, persistently producing 'early-born' γ neurons. Interestingly when combining *Syp-RNAi* with the Mamo transgene, the newborn cells (*Figure 5H'*, cells adjacent to NBs) begin to take on a γ-like fate (expressing Abrupt) before a majority transform into an α'/β'-specific, strong Trio expression pattern and adopt α'/β'-like axon morphology. This suggests that Mamo functions downstream of the temporal fate specification genes, but is capable of overriding downstream signals in α/β and γ neurons to promote α'/β' terminal fate.

## Mamo, a temporally patterned terminal selector gene

What we describe about the BTB-ZF transcription factor, Mamo's role in α'/β' cell fate easily fits into the definition of a terminal selector gene, coined by Oliver Hobert (*Hobert, 2008*). Terminal selector genes are a category of 'master regulatory' transcription factors that control the specific terminal identity features of individual neuronal types (*Hobert, 2016*; *Hobert, 2008*). Key aspects of terminal selector genes are that they are expressed post-mitotically in neurons as they mature and they are continuously expressed (often via autoregulatory mechanisms) to maintain the terminal differentiated state of the neuron. Correspondingly, *mamo* transcription is initiated in newborn, post-mitotic neurons (*Figure 4D*) and Mamo protein expression is visible beginning in young/maturing neurons (*Figure 1C*). After transcription initiation, Mamo positively regulates its own expression (*Figure 4L and M*) and continues to be expressed in α'/β' neurons into adulthood (*Figure 6—figure supplement 1C*). The other quintessential feature of terminal selector genes is that they regulate a battery of terminal differentiation genes, so that removing a terminal selector gene results in a loss of the specific identity features of a neuron type and misexpression can drive those features in other neurons (*Hobert, 2016*; *Hobert, 2008*). Indeed, removing Mamo with RNAi results in the loss of α'/β' identity, both developmentally (*Figure 5C and C', E and E'*) and into adulthood (*Figure 6C and C'*). Further, overexpressing Mamo in either α/β or γ MB neurons results in shift to α'/β' fate (*Figure 5F*

*and F', H and H', Figure 6D and D'*). Individual terminal selectors do not often function alone, but in combination with other terminal selectors. Therefore, there are likely terminal selectors downstream of the MB NB-specific genes that contribute to each of the MB neuron types. In this way, the lineage-specific and temporal patterning programs can combine to define individual neuron types. This feature enables the reutilization of terminal selector genes to create disparate neuron types when used in distinct combinations (*Hobert, 2016*). This further suggests that temporally expressed Mamo serves as a temporally defined terminal selector gene in other lineages, such as the AL lineages we describe here (*Figure 7*).

## Temporal fating mechanism of Chinmo

Altering Chinmo levels via upstream RNA-binding proteins (*Liu et al., 2015*) or miRNAs (*Wu et al., 2012*), or by reducing Chinmo with RNAi (*Figure 5D and D'* and *Figure 5—figure supplement 3*) all result in shifts in the ratio of neurons with different neuronal temporal fates. This evidence suggests a mechanism where Chinmo acts in newborn neurons to promote temporal fate specification. A recent publication suggested that Chinmo affects temporal fate via a neuronal remodeling mechanism by controlling Ecdysone signaling (*Marchetti and Tavosanis, 2017*). As in our first Chinmo study (*Zhu et al., 2006*), Marchetti and Tavosanis demonstrate that Chinmo is required for EcR-B1 expression; however it remains unclear whether Chinmo directly affects EcR-B1 expression or if the Chinmo-dependent EcR-B1 expression is the sole mechanism for γ neuron temporal fate specification. Moreover, neuronal temporal fate is not accurately determined by neuronal morphology alone, particularly when ecdysone signaling has known effects on MB cell morphology (*Lee et al., 2000*) and fate (*Kucherenko et al., 2012*). Ecdysone receptor signaling is highly pleiotropic (*Alyagor et al., 2018*), including ligand-independent functions (*Mouillet et al., 2001*) making dominant-negative and overexpression studies difficult to interpret. Therefore, further investigation is needed to clarify the roles of Ecdysone receptor signaling in MB neuronal temporal fate and remodeling. We hope to address this in a follow-up paper. This current manuscript strongly promotes the idea that Chinmo functions in newborn neurons to promote temporal fate as weak Chinmo expression (*Figure 3—figure supplement 1*) directly precedes Mamo transcription (*Figure 4*) and Mamo is essential for specification (*Figure 5*) and maintenance of α'/β' fate (*Figure 6*).

## Evolutionary conservation

We describe a multilayered hierarchical system to define distinct neuronal temporal fate that culminates in the expression of a terminal selector gene. Analogous mechanisms likely underlie temporal patterning in mammalian brains. However, whether orthologous genes play equivalent roles in mammalian temporal patterning has not been fully investigated. The Imp and Syp RNA-binding proteins are evolutionarily conserved. Both homologs are highly expressed in the developing mouse brain and play vital roles in neural development and/or neuronal morphology (*Chen et al., 2012*; *Mori et al., 2001*; *Perycz et al., 2011*; *Williams et al., 2016*; *Xing et al., 2012*). The opposing functions of Imp and Syp also appear to be conserved, as the murine orthologs IMP1 and SYNCRIP bind the identical RNA to either promote (*Donnelly et al., 2013*) or repress axon growth (*Williams et al., 2016*), respectively. Moreover, IMP1 expression in fetal mouse neural stem cells plays important roles in stem cell maintenance and proper temporal progression of neurogenesis. It would likewise be very interesting to explore SYNCRIP in the context of temporal patterning.

While Chinmo and Mamo have no clear mammalian orthologs, they are both BTB-ZF (broad-complex, tram-track and bric-à-brac - zinc finger) transcription factors (*Mukai et al., 2007*; *Zhu et al., 2006*). The BTB domain is a protein interaction domain that can form homo or heterodimers and also binds transcriptional regulators such as repressors, activators and chromatin remodelers (*Perez-Torrado et al., 2006*). The $C_2H_2$ (Krüppel-like) zinc fingers bind DNA—providing target specificity. BTB-ZF proteins have been found to be critical regulators of developmental processes, including neural development (*Chaharbakhshi and Jemc, 2016*; *Siggs and Beutler, 2012*). Indeed, the BTB-zinc finger protein, Zbtb20, appears to be essential for early-to-late neuronal identity in the mouse cortex (*Tonchev et al., 2016*). Zbtb20 is temporally expressed in cortical progenitors and knockout results in cortical layering defects (*Tonchev et al., 2016*), as the inside-out layering of the cortex follows neuronal birth order. While mutations of other brain-expressed BTB-ZF proteins also show

cortical layering phenotypes (*Carter et al., 2000*; *Okado et al., 2009*), potential roles in temporal patterning have not been explored.

## Conclusions

In this study, we illustrate a fate specification process in which a layered series of temporal protein gradients guide the expression of terminal selector genes. The first-tier temporal gradients are expressed in neural stem cells, followed by a restricted expression window in newborn neurons to finally induce a terminal selector gene in a subset of neurons as they mature. This time-based subdivision of neuronal fate can likely be further partitioned, finally resulting in sequentially born neurons with distinct cell fates. We demonstrate that Mamo, a BTB-ZF transcription factor, delineates α'/β' neurons, the middle temporal window of the MB lineages. Corresponding data in the AL lineages suggest that Mamo may serve as a temporally defined, terminal selector gene in a variety of lineages in the *Drosophila* brain. Mamo expression is regulated transcriptionally by the descending Chinmo BTB-ZF transcription factor gradient and post-transcriptionally by the Syp RNA binding protein. This multi-tiered, bimodal regulation ensures that only the progeny in a precise temporal window (those with both weak Chinmo and significant Syp levels) can effectively activate the terminal selector gene, *mamo*. This discovery attests to the power of gradients in creating diverse cells from a single progenitor. Utilizing layers of temporal gradients to define discrete temporal windows mirrors how in early embryos the spatial gradients of RNA-binding proteins and transcription factors specify the fly's A/P axis. This paradigm provides considerable complexity of gene network regulation, leading to abundant neural cell diversity.

## Materials and methods

### Key resources table

| Reagent type (species) or resource | Designation | Source or reference | Identifiers | Additional information |
|---|---|---|---|---|
| Antibody | anti-Mamo (Rabbit polyclonal) | This paper: Materials and methods | | (1:1000), Lee T, Janelina Research Campus, HHMI |
| Antibody | anti-GFP, Alexa488 (Rabbit polyclonal) | Thermo Fisher Scientific | Cat # A-21311; RRID:AB_221477 | (1:1000) |
| Antibody | anti-GFP (Chicken polyclonal) | Thermo Fisher Scientific | Cat # A10262; RRID:AB_2534023 | (1:1000) |
| Antibody | anti-Chinmo (Rabbit polyclonal) | *Zhu et al., 2006* | | (1:1000) |
| Antibody | anti-Chinmo (Rat) | *Wu et al., 2012* | | (1:500) |
| Antibody | anti-Trio (Rabbit) | *Awasaki et al., 2000* | | (1:1000) |
| Antibody | anti-Abrupt (Rabbit) | *Hu et al., 1995* | | (1:200) |
| Antibody | anti-Imp (Rabbit) | gift from Paul Macdonald | | (1:600) |
| Antibody | anti-Syp (Genia pig) | gift from Ilan Davis | | (1:500) |
| Antibody | anti-Trio (Mouse monoclonal) | Developmental Studies Hybridoma Bank | 9.4A;Registry ID:AB_528494 | (1:200) |
| Antibody | anti-Fas-II (Mouse monoclonal) | Developmental Studies Hybridoma Bank | 1D4; Registry ID:AB_528235 | (1:40) |

*Continued on next page*

*Continued*

| Reagent type (species) or resource | Designation | Source or reference | Identifiers | Additional information |
|---|---|---|---|---|
| Antibody | anti-nc82 (Mouse monoclonal) | Developmental Studies Hybridoma Bank | nc82; Registry ID:AB_2314866 | (1:100) |
| Antibody | anti-chicken, Alexa488 (Goat) | Thermo Fisher Scientific | Cat # A-11039; RRID:AB_2534096 | (1;500) |
| Antibody | anti-mouse, Alexa568 (Goat) | Thermo Fisher Scientific | Cat # A-11031; RRID:AB_144696 | (1;500) |
| Antibody | anti-rabbit, Alexa647 (Goat) | Thermo Fisher Scientific | Cat # A-21244; RRID:AB_2535812 | (1;500) |
| Antibody | anti-Rat, Alexa568 (Goat) | Thermo Fisher Scientific | Cat # A-11077; RRID:AB_2534121 | (1;500) |
| Antibody | anti-rabbit, Alexa568 (Goat) | Thermo Fisher Scientific | Cat # A-11036; RRID:AB_10563566 | (1;500) |
| Antibody | anti-mouse, Alexa647 (Donkey) | Jackson Immuno Research lab, Inc. | Cat # 715-605-151 | (1;500) |
| Antibody | anti-Rat, DyLight405 (Goat) | Jackson Immuno Research lab, Inc. | Cat # 112-475-167 | (1;200) |
| Chemical compound, drug | Paraformadehyde 20% Solution, EM Grade | Electron Microscopy Sciences | Cat # 15713 | |
| Chemical compound, drug | Phosphate Buffered Saline 10X, Molecular Biology Grade | Thermo Fisher Scientific | Cat # 46–013 CM | |
| Chemical compound, drug | Triton X-100 | Sigma-Aldrich | Cat # 329830772 | |
| Chemical compound, drug | SlowFadeTM Gold antifade Mountant | Thermo Fisher Scientific | Cat # S36936 | |
| Chemical compound, drug | RNase-free 1x PBS | Thermo Fisher Scientific | Cat # BP2438-4 | |
| Chemical compound, drug | Acetic Acid, Glacial | Thermo Fisher Scientific | Cat # A38S-500 | |
| Chemical compound, drug | Sodium borohydride | Acros Organics/ Thermo Fisher Scientific | Cat # AC448481000 | |
| Chemical compound, drug | Invitrogen SSC (20X) | Thermo Fisher Scientific | Cat # AM9763 | |
| Chemical compound, drug | Hi-Di formamide | Applied Biosystems/ Thermo Fisher Scientific | Cat # 4311320 | |
| Chemical compound, drug | Alfa Aesar Denhardt's solution (50X) | Alfa Aesar/Thermo Fisher Scientific | Cat # AAJ63135AD | |
| Chemical compound, drug | tRNA from Baker's yeast | Roche | Cat # 10109495001 | |
| Chemical compound, drug | UltraPure Salmon Sperm DNA Solution | Thermo Fisher Scientific | Cat # 15632011 | |
| Chemical compound, drug | Corning 10% SDS | Corning/Thermo Fisher Scientific | Cat # 46–040 CI | |
| Chemical compound, drug | Deionized formamide | Ambion/Thermo Fisher Scientific | Cat # AM9342 | |
| Chemical compound, drug | RNase*Zap* RNase Decontamination Solution | Thermo Fisher Scientific | Cat # AM9780 | |

*Continued on next page*

*Continued*

| Reagent type (species) or resource | Designation | Source or reference | Identifiers | Additional information |
|---|---|---|---|---|
| Chemical compound, drug | Poly-L-lysine hydrobromide | Sigma-Aldrich | Cat # P1524-25MG | |
| Chemical compound, drug | Cy3 Mono-Reactive Dye Pack | GE Healthcare Life Sciences | Cat # PA23001 | |
| Chemical compound, drug | Cy5 Mono-Reactive Dye Pack | GE Healthcare Life Sciences | Cat # PA25001 | |
| Chemical compound, drug | Ethyl alcohol, pure | Sigma-Aldrich | Cat # 459844 | |
| Chemical compound, drug | Xylenes | Thermo Fisher Scientific | Cat # X5-500 | |
| Chemical compound, drug | DPX mountant | Electron Microscopy Sciences | Cat # 13512 | |
| Genetic reagent (*D. melanogaster*) | *tub-Gal80ts* | Bloomington *Drosophila* stock center | BDSC:7018; FLYB:FBst0007018; RRID:BDSC_7018 | FlyBase symbol:P{w[+mC]=tubP-GAL80[ts]}ncd[GAL80ts-7] |
| Genetic reagent (*D. melanogaster*) | *UAS-Syp-RNAi* | Vienna *Drosophila* RNAi Center | VDRC:v33012; FLYB:FBst0459886 | FlyBase symbol: P{GD9477}v33012 |
| Genetic reagent (*D. melanogaster*) | *UAS-mamo-RNAi* | Bloomington *Drosophila* stock center | BDSC:51770; FBti0157732; RRID:BDSC_51770 | FlyBase symbol: P{TRiP.HMC03325}attP40 |
| Genetic reagent (*D. melanogaster*) | *UAS-mamo-RNAi* | Bloomington *Drosophila* stock center | BDSC: 44103; FBti0158705; RRID:BDSC_44103 | FlyBase symbol: P{TRiP.HMS02823}attP40 |
| Genetic reagent (*D. melanogaster*) | *UAS-mCD8-GFP; +; GAL4-OK107* | *Connolly et al., 1996* | | |
| Genetic reagent (*D. melanogaster*) | *UAS-chinmo-RNAi* | *Liu et al., 2015* | | |
| Genetic reagent (*D. melanogaster*) | *UAS-chinmo-GOF (UAS-chinmo-3UTR)* | *Zhu et al., 2006* | | |
| Genetic reagent (*D. melanogaster*) | *UAS-Syp-GOF* | *Liu et al., 2015* | | |
| Genetic reagent (*D. melanogaster*) | *UAS-mamo-3ZFs-GOF* | *Mukai et al., 2007* | | |
| Genetic reagent (*D. melanogaster*) | *UAS-mamo-4ZFs-GOF* | This paper: Materials and methods | | Lee T, Janelina Research Campus, HHMI |
| Genetic reagent (*D. melanogaster*) | *UAS-mamo-5ZFs-GOF* | This paper: Materials and methods | | Lee T, Janelina Research Campus, HHMI |
| Genetic reagent (*D. melanogaster*) | *Dpn > KDRT-stop-KDRT >Cre PEST; act > loxP-stop-loxP> LexA::P65, lexAop2-myr::GFP; GR44F03-KD* | *Awasaki et al., 2014* | | |
| Genetic reagent (*D. melanogaster*) | *LexAop2-chinmo-RNAi* | This paper: Materials and methods | | Lee T, Janelina Research Campus, HHMI |
| Genetic reagent (*D. melanogaster*) | *LexAop2-Syp-RNAi* | *Ren et al., 2017* | | |
| Genetic reagent (*D. melanogaster*) | *UAS-mCD8-GFP-insu-UAS-rCD2-RNAi, chinmo1, FRT40A* | *Kao et al., 2012* | | |
| Genetic reagent (*D. melanogaster*) | *hs-FLPop; tub-GAL80, FRT40A; +; GAL4-OK107* | This paper: Materials and methods | | Lee T, Janelina Research Campus, HHMI |

*Continued on next page*

*Continued*

| Reagent type (species) or resource | Designation | Source or reference | Identifiers | Additional information |
|---|---|---|---|---|
| Software, algorithm | Fiji | NIH; *Schindelin et al., 2012* | https://fiji.sc/ | |
| Software, algorithm | Adobe Photoshop | Adobe Systems, San Jose, CA | https://www.adobe.com/products/photoshop.html | |
| Software, algorithm | Adobe Illustrator | Adobe Systems, San Jose, CA | https://www.adobe.com/products/illustrator.html | |
| Software, algorithm | Python | Python Software Foundation | https://www.python.org/ | |
| Software, algorithm | Flybase 2.0 | *Thurmond et al., 2019* | http://flybase.org | |
| Software, algorithm | Matplotlib | *Hunter, 2007* | https://matplotlib.org | |

All strains of the *Drosophila melanogaster* used in this study were listed below. Stocks were raised at 25°C incubator.

For labeling entire MB lineages, we used *UAS-mCD8-GFP; +; GAL4-OK107* (*Connolly et al., 1996*). For temperature shift assay, we used a temperature sensitive GAL80 (*McGuire et al., 2003*). The following transgenenic flies were used. (1) *UAS-Syp-RNAi* (stock# 33012, VDRC stock center), (2) *UAS-mamo-RNAi* (stock# 51770 and # 44103, Bloomington stock center), (3) *UAS-chinmo-RNAi* (*Liu et al., 2015*), (4) *UAS-chinmo-GOF* (*UAS-chinmo-3UTR*; *Zhu et al., 2006*), (5) *UAS-Syp-GOF* (*Liu et al., 2015*), (6) *UAS-mamo-3ZFs-GOF* (*Mukai et al., 2007*), (7) *UAS-mamo-4ZFs-GOF* (this study), (8) *UAS-mamo-5ZFs-GOF* (this study).

For labeling AL lineages, we used *Dpn > KDRT-stop-KDRT>Cre PEST; act >loxP-stop-loxP>LexA::P65, lexAop2-myr::GFP; GR44F03-KD* (*Awasaki et al., 2014*). The following transgenic flies were used. (1) *LexAop2-chinmo-RNAi* (this work) (2) *LexAop2-Syp-RNAi* (*Ren et al., 2017*).

To generate *chinmo* mutant MARCM clones, *UAS-mCD8GFP-insu-UAS-rCD2-RNAi, chinmo[1], FRT40A* flies (*Kao et al., 2012*) were crossed with *hs-FLPop; tub-GAL80, FRT40A; +; GAL4-OK107* flies. The crossed flies laid eggs in the vials for every four hours. MARCM clones were induced at newly hatched larvae (NHL) via heat shock at 37°C for 30 mins and dissected at 84 hr ALH.

To express *mamo-GOF (3ZFs, 4ZFs, 5ZFs)*, we made *UAS-mamo-GOF (4ZFs, 5ZFs)* flies. The driver *GAL4-OK107* was utilized for driver dependent ectopic induction of the isoform *mamo-3ZFs*, *mamo-4ZFs*, and *mamo-5ZFs*.

## Temporal induction of RNAi and Overexpression after adult eclosion

Embryo with genotype *UAS-mamo-RNAi/UAS-GFP; tub-GAL80ts/OK107 Gal4* or *UAS-mamo-GOF/UAS-GFP; tub-GAL80ts/OK107 Gal4* were cultured at 18°C until adult eclosion. The Adult were incubated at 29°C to inactivate the temperature-sensitive GAL80 and cultured for 7 days or 21 days. The Adult were dissected right after the culture.

## Temporal induction of Overexpression after pupal stage

Embryo with genotype *UAS-mamo-GOF/UAS-GFP; tub-GAL80ts/OK107 Gal4* were cultured at 18°C until white pupae. The white pupae were collected and cultured at 18°C for 2 days. Then, they were incubated at 29°C to inactivate the temperature- sensitive GAL80 and cultured for 7 days. The Adult were dissected right after the culture.

## Antibodies and immunostaining

Fly brains at specific developmental stages were dissected in the 1X Phosphate Buffered Saline (PBS, Thermo Fisher Scientific). After brains were fixed in 4% paraformaldehyde (Electron Microscopy Sciences) for 35mins, they were wash in 0.5% PBT (1X PBS with 0.5% Trioton X-100, Sigma-Aldrich) for three times and immunostained for primary antibodies for overnight as described previously (*Lin et al., 2012*). The brains were washed in 0.5% PBT and immunostained for secondary antibodies for overnight. The next day, the brains were washed and mounted in SlowFade Gold

antifade Mountant (Thermo Fisher Scientific). The following primary antibodies were used: chicken anti-GFP (1:1000, A10262, Life Technologies), rabbit anti-Mamo (1:1000, this study), rabbit anti-Chinmo (1:1000, *Zhu et al., 2006*), rat anti-Chinmo (1:500, *Wu et al., 2012*), rabbit anti-Trio (1:1000, (*Awasaki et al., 2000*), mouse anti-Trio (1:200, 9.4A, Developmental Studies Hybridoma Bank), mouse anti-Fas-II (1:40, 1D4, Developmental Studies Hybridoma Bank), rabbit anti-Abrupt (1:200, *Hu et al., 1995*), Rabbit anti-Imp (1:600, gift from Paul Macdonald, University of Texas at Austin) and Genia pig anti-Syp (1:500, gift from Ilan Davis, University of Oxford). All corresponding fluorescent secondary antibodies (1:500) were purchased from Life Technologies. Images of whole-mount fly brains were acquired using a Zeiss LSM 710 or LSM 880 confocal microscope and processed with Fiji-Image J and Adobe Photoshop.

## Antibody generation

The polyclonal rabbit anti-Mamo antibody was raised against the QKREASDRSSPTPAC peptide (aa 273 to aa 286 in Mamo, GenScript).

## Molecular biology

To generate miRNA construct for *chinmo*, two polycistronic transcripts that each encoding two miR-NAs against *chinmo*. The miRNA targeting sequences were 5'- ACAGAGATACGGACAAAGATAC-3' and 5'-CATCTACCGGCCTATTAACTAC-3'. The above transcripts were inserted after the lexAop promoter in the pMLH Plasmid (*Pfeiffer et al., 2010*). The restriction enzyme sites used were 5'-NotI to 3'-XhoI.

To generate different isoforms of *mamo*, full-length DNA sequence of *mamo-4ZFs* and C-terminal sequence of *mamo-5ZFs* were obtained from flybase. Two DNA fragments were synthesized by GenScript. The fragment of *mamo-4ZFs* was flanked by XhoI and XbaI. To generate the fragment of *mamo-5ZFs*, the C-terminus of *mamo-4ZFs* was replaced by the C-terminus of *mamo-5ZFs*, which was following a single cut with Bgl-II. Both full-length fragments of *mamo-4ZFs* and *mamo-5ZFs* were insert into XhoI/XbaI site to replace B3::PEST in pJFRC157-20XUAS-IVS-B3::PEST vector (Addgene plasmid 32136).

The plasmid of *LexAop2-chinmo-RNAi*, *UAS-mamo-4ZFs-GOF* and *UAS-mamo-5ZFs-GOF* was injected by Rainbow Transgenic Flies, Inc (Camarillo,CA,USA). The complete nucleotide sequences of the plasmids will be provided upon request.

## smFISH

The detailed FISH methods, probe design and labeling protocols were as described previously (*Long et al., 2017*). FISH probe sequences for *mamo* nascent and mature transcripts are listed in *Figure 4—source data 2*. The amino modified FISH probes targeting nascent and mature *mamo* transcripts were coupled to Cy3 and Cy5 fluorophores through N-hydroxysuccinimide esters. Fly brains at specific developmental stages were dissected in 1xPBS and fixed in 4% paraformaldehyde at room temperature for 35 min. Tissues were washed in 0.5% PBT, dehydrated, and stored in 100% ethanol at 4℃ overnight. After rehydration in the following day, tissues were incubated in 5% acetic acid at 4℃ for 5 min and fixed in 2% paraformaldehyde in for 35 min at 25℃. Fly brains were then washed in 1 × PBS containing 1% of NaBH4 at 4℃ for 30 min. After a 2 hr incubation in prehybridization buffer (15% formamide, 2 × SSC, 0.1% Triton X-100) at 50℃, fly brains were introduced to hybridization buffer (10% formamide, 2 × SSC, 5 × Denhardt's solution, 1 mg/ml yeast tRNA, 100 µg/ml, salmon sperm DNA, 0.1% SDS) with FISH probes, and incubation at 50℃ for 10 hr and then at 37℃ for an additional 10 hr. Fly brains were washed in a series of washing solutions, dehydrated, cleared in xylene, and mounted in DPX. The confocal images were collected using Zeiss LSM 880 and processed with Fiji-Image J and Adobe Photoshop after the tissues were cured for 24 hr.

## Image analysis

To measure Chinmo, Mamo, Imp, Syp signal intensity in the MB (*Figures 1*, *2* and *3*, *Figure 3—figure supplement 1*, *Figure 4*, *Figure 5—figure supplement 2*), the NBs, newborn neurons and maturing neurons were labeled with OK107-Gal4 > GFP. The definition of newborn and maturing neurons was based on the GFP intensity as described in the *Figure 1* legend. We selectively analyzed white and yellow outlines at the chosen focal planes near the NB region. For NB, we selectively

analyzed those NBs (circled) with a maximum diameter. To compare Chinmo (*Figure 1*, *Figure 3—figure supplement 1* and *Figure 5—figure supplement 2*), Mamo (*Figures 1*, *2* and *3* and *Figure 4*) and Imp/Syp (*Figure 5—figure supplement 2*) levels across various genotypes, the samples were immunostained simultaneously for every single experiment. The images were taken using the same confocal setting (pinhole size, gain, laser power, etc.) and an image of selected focal plane was exported to Adobe Photoshop. A hand-drawn mask was created for the newborn neurons (for Chinmo), the maturing neurons (for Mamo) and the cytoplasmic (for Imp/Syp) region of interest at selected focal plane. The averaged grayscale value for each pre-defined region was calculated using the 'Histograms' algorithm in Photoshop. The grayscale values of Chinmo and Mamo and Imp/Syp were normalized to the background staining in the developing central brains.

## Genomic

The mamo isoform transcription start site (TSS) and exons are extracted from Flybase annotation (dmel-all-r6.17.gtf). Protein domain information is obtained from uniprot (http://www.uniprot.org; Q9VY72 for RH, RI; M9NEG1 for RF, RG; M9PJM9 for RD, RE; H1UUK0 for RC isoforms). In house custom program in Python (http://www.python.org) with matplotlib (http://www.matplotlib.org) library is used to make the gene structure diagram.

## Quantification and statistical analysis

Quantification of Mamo positive neurons in fly brains in *Figure 2* was analyzed with Student's t test. Sample size and *P values* are mentioned within the figure legend and *Figure 2—source data 1*.

## Data and software availability

Customized MATLAB and Python scripts used in this paper are in the *Source code 1*.

# Acknowledgements

We thank NS Sokol and ST Crews and Paul Macdonald and Ilan Davis for sharing antibodies. We thank T Awasaki for sharing antibodies and fly stocks. We thank the Janelia Fly Core, the Bloomington *Drosophila* Stock Center, the Vienna Resource Center (VDRC), the Rainbow transgenic fly, Inc, and GenScript for technical support. We thank C Di Pietro and K Miller for administrative support.

# Additional information

## Competing interests

Robert H Singer: Reviewing editor, *eLife*. The other authors declare that no competing interests exist.

## Funding

| Funder | Grant reference number | Author |
| --- | --- | --- |
| Howard Hughes Medical Institute | | Tzumin Lee |
| National Institute of Neurological Disorders and Stroke | NS083085 | Robert H Singer |

The funders had no role in study design, data collection and interpretation, or the decision to submit the work for publication.

## Author contributions

Ling-Yu Liu, Conceptualization, Data curation, Formal analysis, Validation, Investigation, Visualization, Methodology, Writing—original draft, Writing—review and editing; Xi Long, Investigation, Methodology, Writing—original draft; Ching-Po Yang, Conceptualization, Data curation, Investigation, Visualization, Methodology, Writing—original draft; Rosa L Miyares, Visualization, Writing—original draft; Ken Sugino, Software; Robert H Singer, Supervision, Funding

acquisition; Tzumin Lee, Conceptualization, Supervision, Funding acquisition, Investigation, Visualization, Methodology, Writing—original draft, Project administration, Writing—review and editing

## Author ORCIDs
Ling-Yu Liu [ID] https://orcid.org/0000-0002-7096-4150
Xi Long [ID] http://orcid.org/0000-0002-0268-8641
Ken Sugino [ID] http://orcid.org/0000-0002-5795-0635
Robert H Singer [ID] http://orcid.org/0000-0002-6725-0093
Tzumin Lee [ID] https://orcid.org/0000-0003-0569-0111

## Decision letter and Author response
Decision letter https://doi.org/10.7554/eLife.48056.033
Author response https://doi.org/10.7554/eLife.48056.034

## Additional files

### Supplementary files
• Source code 1. The code for mamo gene structure diagram.
DOI: https://doi.org/10.7554/eLife.48056.030

• Transparent reporting form
DOI: https://doi.org/10.7554/eLife.48056.031

### Data availability
All data generated or analyzed during this study are included in the manuscript and supporting files. Source data files have been provided for all figures.

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
