## [Decision Letter]

Thank you for submitting your article "Mamo decodes hierarchical temporal gradients into terminal neuronal fate" for consideration by *eLife*. Your article has been reviewed by two peer reviewers, and the evaluation has been overseen by Oliver Hobert as the Reviewing Editor and K VijayRaghavan as the Senior Editor. Chris Doe has agreed to reveal his identity as one of the reviewers.

The reviewers have discussed the reviews with one another and the Reviewing Editor has drafted this decision to help you prepare a revised submission.

This manuscript describes Mamo as an interesting new factor that controls the ordered birth of different neuron types during neurogenesis. The fact that it is expressed in both the mushroom body and in antennal lobe lineages further suggests that it is a general temporal factor and not lineage specific. It is therefore a significant finding that deserves to be published. However, prior to publication, the authors need to (a) extend their analysis (with some very simple and doable experiments) and (b) clarify a number of points.

The two set of experiments required to deepen their analysis are:

1) Remove Mamo from adults and assay Kenyon cell fate and function. Again, technically trivial (OK107-Gal4, Gal80.ts, *UAS-mamo-RNAi*) and would help solidify the conclusion that Mamo is a terminal selector gene. Similarly, misexpress Mamo in specifically in adults (OK107-Gal4, Gal80.ts, *UAS-mamo-GOF*) and assay for ectopic a'/b' neurons.

2) Overexpress constitutive Mamo (replace UTRs) and assay for Imp, Syp, and Chinmo. All three should have normal expression patterns if the hierarchical regulation model is true. This is technically trivial, and the *mamo* construct is even mentioned in the Discussion (although incorrectly cited as being in Figure 4L, which is a different experiment).

In addition, the following additional points need to be clarified:

I) They need to quantity the number of neurons present/missing in the different genetic backgrounds. This can be done either using the markers the authors describe in the manuscript, or using cell-type specific Gal4/LexA lines. A general question that arises throughout the manuscript is to address the fate of neurons born in different temporal windows when Chinmo, Mamo, Imp or Syp are missing.

II) They also need to clearly indicate in all figures newborn vs. maturing vs. mature neurons. An outline around the different neuron populations (i.e., newborn vs. mature) would make it easier to understand the different expression domains.

III) Finally, the authors need to quantify fluorescence intensity values for Chinmo and Mamo in control and all mutant backgrounds.

In more detail:

- In Figure 1, the authors use OK107-Gal4 to label mushroom body neurons. According to Figure 1A, this driver labels more than 1 neuron type in the mushroom bodies after 72 hrs. Thus, it is not possible to determine whether the Mamo positive staining in 1C is in γ neurons or in α'β' neurons, particularly since maturing vs. newborn neurons is not clearly defined. The authors need to do the same staining using either a γ-Gal4 line (e.g., 201y-Gal4 or 71G10-Gal4) or an α’β’-Gal4 line.

- Similar to the above comment, the authors need to determine which mushroom body cells are Mamo positive in Figure 1D. At this stage, all three mushroom neuron types are present. Do all neuron types eventually become Mamo positive? For example, since Chinmo is not maintained, does Mamo become activated in all cells that once expressed Chinmo once Chinmo levels decline (as shown in Figure 1D)? If so, it would help validate part of the authors model that low Chinmo activates Mamo. The authors need to use Gal4 lines that label the different mushroom body neuron types and stain/quantify Mamo and Chinmo levels during development.

- Chinmo and Mamo levels should be quantified in newborn/maturing/mature neurons through time in Figure 1B-D.

- Quantify Chinmo levels at 84 ALH in Figure 2. It is not enough to place stars in the cartoon, especially since the authors want to show that low Chinmo activates Mamo. At 48h ALH, in Figure 3—figure supplement 1D, Chinmo levels already look undetectable. This is a major point of the paper that should be better characterized/quantified.

- In Figure 2D, the authors need to define what the additional Mamo positive neurons are within the clones. Are they γ neurons (they have low Chinmo, and so they might activate Mamo), or are they additional α’β’ neurons specified earlier? The same is true for Figure 2F. Given the context of the manuscript, it is assumed that they are additional α’β’ neurons but the authors need to demonstrate this point and state clearly their conclusion.

- In Figure 3D, Mamo expression is lost at 84 ALH. However, the authors have previously reported (Lee, Science) that overexpression of Imp or Syp leads to additional α’β’ neurons in the adult, while later in this manuscript, they demonstrate that Mamo is necessary for α’β’ identity. Thus, the authors should put their new findings in context of their previous results. When are additional α’β’ neurons born according to their current model? In the pupal window?

- For Figure 4, the authors need to quantify the smFISH results. For example, it is difficult to know if there is a difference in mature transcript abundance between 4M and 4H. They must show the smFISH results in a Syp-RNAi background alone (They do show Syp-RNAi with Chinmo RNAi).

- Figure 5H: Syp-RNAi with *Mamo-GOF* results in a mushroom body with a majority of α’β’ neurons. However, throughout the manuscript, the authors have made a point to indicate that Mamo requires post-transcriptional regulation by Syp. Thus, this result seems to contradict the authors' earlier results as *Mamo-GOF* should not lead to additional α’β’ neurons in the absence of Syp. This apparent discrepancy must be addressed.

- The authors must quantify the number of α’β’ neurons in the different genetic backgrounds in Figure 5 (e.g. using strong Trio expression of more than one marker.

---

## [Author Response]

This manuscript describes Mamo as an interesting new factor that controls the ordered birth of different neuron types during neurogenesis. The fact that it is expressed in both the mushroom body and in antennal lobe lineages further suggests that it is a general temporal factor and not lineage specific. It is therefore a significant finding that deserves to be published. However, prior to publication, the authors need to (a) extend their analysis (with some very simple and doable experiments) and (b) clarify a number of points.The two set of experiments required to deepen their analysis are:1) Remove Mamo from adults and assay Kenyon cell fate and function. Again, technically trivial (OK107-Gal4, Gal80.ts, UAS-mamo-RNAi) and would help solidify the conclusion that Mamo is a terminal selector gene.

We have included a new Figure (Figure 6), supplementary figures (Figure 6—figure supplements 1 and 2) and a new section in the Results (Mamo maintains α’/β’ cell-fate) to help address Mamo’s role in maintenance of α’/β’ neuronal fate. *Mamo-RNAi* induction in adult mushroom body neurons caused a progressive reduction in Trio expression. *Mamo-RNAi* was very slow to reduce Mamo levels; after one week of expressing *mamo-RNAi*, Mamo protein still remained. This may have been due to strong positive autoregulation of Mamo expression. After three weeks of RNAi, Mamo protein was no longer detectable and we see a substantial reduction in Trio staining, in that there was a complete loss of neurons with α’/β’ characteristic gene expression (Abrupt negative cells expressing Trio in both the cytoplasm and plasma membrane; Trio^PM,Cyto^/Abrupt^-^). The slow action of *mamo-*RNAi and the progressive reduction of Trio staining, suggests that with a longer RNAi induction, Trio would disappear completely.

Similarly, misexpress Mamo in specifically in adults (OK107-Gal4, Gal80.ts, UAS-mamo-GOF) and assay for ectopic a'/b' neurons.

This data is also included in Figure 6 and Figure 6—figure supplements 1 and 2 and a new section in the Results (Mamo stimulates α’/β’ specific gene expression in mature MB neurons). As predicted, overexpressing Mamo in adult MBs also caused a significant (p<0.01) shift to produce more neurons with α’/β’ characteristic gene expression (Trio^PM,Cyto^/Abrupt-), from 22% in the control to 42% with *mamo-GOF*.

2) Overexpress constitutive Mamo (replace UTRs) and assay for Imp, Syp, and Chinmo. All three should have normal expression patterns if the hierarchical regulation model is true. This is technically trivial, and the mamo construct is even mentioned in the Discussion (although incorrectly cited as being in Figure 4L, which is a different experiment).

We have included in a new supplementary figure (Figure 5—figure supplement 2) to address this point.

The Results section reads:

”…either the *mamo-RF* or *mamo-RG* isoform lies downstream of Imp/Syp gradients and Chinmo in α’/β’ temporal fate determination. In further support of a hierarchy, 4ZF Mamo overexpression does not alter Imp, Syp or Chinmo levels (Figure 5—figure supplement 2).”

We have corrected the citation.

In addition, the following additional points need to be clarified:I) They need to quantity the number of neurons present/missing in the different genetic backgrounds. This can be done either using the markers the authors describe in the manuscript, or using cell-type specific Gal4/LexA lines. A general question that arises throughout the manuscript is to address the fate of neurons born in different temporal windows when Chinmo, Mamo, Imp or Syp are missing.

As the neuronal fate markers are not distinguishable during larval stages, it is difficult to assess the neuronal fate until the late pupa or adult stage. Therefore, we have addressed the neuronal fate in adult stages in Figure 5. We include quantifications for all three neuron populations for most genotypes (Figure 5B, D, F and Figure 5—figure supplement 3).

II) They also need to clearly indicate in all figures newborn vs. maturing vs. mature neurons. An outline around the different neuron populations (i.e., newborn vs. mature) would make it easier to understand the different expression domains.

Newborn and young/maturing neurons are identified in Figures 1-4. Newborn neurons are identified by the very dim GFP expression as described in Zhu et al., 2006. Young/maturing neurons lie immediately adjacent to the newborn neurons with a slightly higher GFP intensity than newborn neurons.

III) Finally, the authors need to quantify fluorescence intensity values for Chinmo and Mamo in control and all mutant backgrounds.

Quantifications are now included in Figure 1—figure supplement 1, Figure 2—figure supplement 1, Figure 3—figure supplement 1, Figure 3—figure supplement 2, and Figure 4—figure supplement 1.

In more detail:- In Figure 1, the authors use OK107-Gal4 to label mushroom body neurons. According to Figure 1A, this driver labels more than 1 neuron type in the mushroom bodies after 72 hrs. Thus, it is not possible to determine whether the Mamo positive staining in 1C is in γ neurons or in α’β’ neurons, particularly since maturing vs. newborn neurons is not clearly defined. The authors need to do the same staining using either a γ-Gal4 line (e.g., 201y-Gal4 or 71G10-Gal4) or an α’β’-Gal4 line.

We have now defined the newborn and young/maturing neuronal populations throughout Figure 1. In 1C, Mamo positive cells are confined to the young/maturing neuronal population. A new figure (Figure 1—figure supplement 1B) further addresses that γ neurons (marked by 201y-Gal4) are negative for Mamo at 84h ALH and Figure 1—figure supplement 1C shows the onset of γ neuron expression of Mamo at 0h APF.

The text reads:

“To validate that at 84h ALH, Mamo is in fact expressed in prospective α’/β’ rather than γ neurons, we used a γ neuron-specific driver and confirmed that there is no overlap with Mamo expression (Figure 1—figure supplement 1B).”

- Similar to the above comment, the authors need to determine which mushroom body cells are Mamo positive in Figure 1D. At this stage, all three mushroom neuron types are present. Do all neuron types eventually become Mamo positive? For example, since Chinmo is not maintained, does Mamo become activated in all cells that once expressed Chinmo once Chinmo levels decline (as shown in Figure 1D)? If so, it would help validate part of the authors model that low Chinmo activates Mamo. The authors need to use Gal4 lines that label the different mushroom body neuron types and stain/quantify Mamo and Chinmo levels during development.

Please see the point above to address some aspects.

In Figure 1D, both γ and α’/β’ neurons are Mamo positive. We now show in Figure 1—figure supplement 1C that at 0h APF γ neurons expressed Mamo in addition to a second population (α’/β’). In addition, we have included a new figure (Figure 6—figure supplement 1C) to address Mamo expression in the adult MB. Mamo is expressed in γ and α’/β’ neurons (identified by Trio^PM^ and Trio^PM,Cyto^). It is, however, not clear whether the γ neuron induction of Mamo is related to Chinmo levels.

Text in the Results section:

“γ neurons, which express high Chinmo in early larval stages, begin to express Mamo during puparium formation (Figure 1—figure supplement 1C).”

Text in the Discussion:

“Transcription initiation is not the only requirement for Mamo protein expression; Syp is also required (Figures 3 and 4, discussed below). […] It has not yet been tested whether weak Chinmo levels are required for later Mamo expression in γ neurons. It is therefore possible that Mamo expression is controlled at this stage by an additional factor(s).”

- Chinmo and Mamo levels should be quantified in newborn/maturing/mature neurons through time in Figure 1B-D.

We have included quantification in a new figure (Figure 1—figure supplement 1A).

- Quantify Chinmo levels at 84 ALH in Figure 2. It is not enough to place stars in the cartoon, especially since the authors want to show that low Chinmo activates Mamo. At 48h ALH, in Figure 3—figure supplement 1D, Chinmo levels already look undetectable. This is a major point of the paper that should be better characterized/quantified.

We have included a new quantification result in Figure 3—figure supplement 1G. We agree that the intensity of Chinmo immunostaining is difficult to visualize. Weak Chinmo levels are hard to see, but immunofluorescent quantification helps us understand how modifying genotype can affect Chinmo levels. We see low Chinmo levels in any genetic manipulation expected to cause a Chinmo reduction.

- In Figure 2D, the authors need to define what the additional Mamo positive neurons are within the clones. Are they γ neurons (they have low Chinmo, and so they might activate Mamo), or are they additional α’β’ neurons specified earlier? The same is true for Figure 2F. Given the context of the manuscript, it is assumed that they are additional α’β’ neurons but the authors need to demonstrate this point and state clearly their conclusion.

We wish we could easily address cell fate of Mamo positive cells in *chinmo-*RNAi expressing animals during larval stages (Figures 2D and F). However, due to lack of larval cell-type markers, we instead wait until adult stages when cell fate can be better distinguished (Figure 5D and D’). We have now provided the percentages of adult MB neuron fates based on Trio and Abrupt markers (Figure 5—figure supplement 3).

*Chinmo-RNAi* causes a decrease in the percent of γ neurons, the percent of α’/β’ neurons did not change considerably from control animals, but the percent of α/β neurons expanded significantly.

Clearly, the prematurely expressing Mamo cells do not become γ neurons, as γ neurons decreased from ~40% in control animals to ~20% with *chinmo-RNAi*. This is supported by additional findings that γ neurons do not turn on Mamo until they undergo remodeling around puparium formation (Figure 1—figure supplement 1B and C). We believe this is due to lack of Syp expression in γ neurons. We have additional data that will be included in a subsequent paper regarding the cause of this onset of Mamo in γ neurons. (Liu et al., in preparation). However, as of this time, we have not tested whether weak Chinmo is also required for pupal onset of Mamo. We now include discussion of this point (subsection “Weak Chinmo on Mamo protein expression”).

Together, this suggests that the weak Chinmo expression window and subsequent Mamo expression window shifted earlier rather than being extended. The Chinmo expression levels with *chinmo-RNAi* were very low (Figure 3—figure supplement 1G, compare 72h ALH control with 48h *chinmo-RNAi*) and any subsequent reduction (by Syp post-transcriptional regulation) would likely result in a Chinmo-off window, and thus α/β neurons.

The Results now state:

“With premature weak Chinmo (*chinmo-RNAi),* g neuron production significantly (p<0.001) decreased from 38 ± 0.3% in control animals to 18 ± 1.6% with *chinmo-RNAi* (Figure 5D and D’ and Figure 5—figure supplement 3). […] The percentage of α/β characteristic Abrubtp^-^/Trio^-^ cells increased from 41 ± 0.5% in control to 64 ± 1.1% with *chinmo-RNAi*
(p<0.001).”

- In Figure 3D, Mamo expression is lost at 84 ALH. However, the authors have previously reported (Lee, Science) that overexpression of Imp or Syp leads to additional α’β’ neurons in the adult, while later in this manuscript, they demonstrate that Mamo is necessary for α’β’ identity. Thus, the authors should put their new findings in context of their previous results. When are additional α’β’ neurons born according to their current model? In the pupal window?

In order to improve flow/understanding and provide a clearer justification of why we examined Syp’s role in Mamo expression, we decided to remove Imp data from Figure 3. We felt that it distracted from, rather than clarified our main points.

However, to address your question, in Imp overexpression, Chinmo is still expressed in newborn neurons at pupal stages (a time when Chinmo is normally off). We therefore believe that additional α’/β’ neurons are generated in the pupal window.

- For Figure 4, the authors need to quantify the smFISH results. For example, it is difficult to know if there is a difference in mature transcript abundance between 4M and 4H. They must show the smFISH results in a Syp-RNAi background alone (They do show Syp-RNAi with Chinmo RNAi).

We have included quantification for *mamo* mature transcripts in Figure 4—figure supplement 1. As there is no Mamo protein produced with Syp-RNAi, we do not feel that the smFISH experiment is essential. Chinmo-RNAi servers as a control for Chinmo-RNAi with Syp-RNAi.

- Figure 5H: Syp-RNAi with Mamo-GOF results in a mushroom body with a majority of α’β’ neurons. However, throughout the manuscript, the authors have made a point to indicate that Mamo requires post-transcriptional regulation by Syp. Thus, this result seems to contradict the authors' earlier results as Mamo-GOF should not lead to additional α’β’ neurons in the absence of Syp. This apparent discrepancy must be addressed.

We apologize for any misunderstanding. The *Mamo-GOF* experiments utilize a transgene that does not have UTRs or introns and thus is not subject to Syp post-transcriptional regulation. We have now made this clear in the text when recounting the testing of Mamo transgenes and when describing Figure 5H.

- The authors must quantify the number of α’β’ neurons in the different genetic backgrounds in Figure 5 (e.g. using strong Trio expression of more than one marker.

We have quantified cell type percentages for Control animals, *chinmo-RNAi* and Mamo overexpression geneotypes in Figure 5—figure supplement 3. *mamo-RNAi, chinmo-RNAi plus mamo-RNAi, and syp-RNAi* all lack α’/β’ neurons, shifting to nearly all α/β or γ neurons so quantifications were not crucial to understand how cell fate was altered.